# Reviewing the Impact of Powder Cohesion on Continuous Direct Compression (CDC) Performance

**DOI:** 10.3390/pharmaceutics15061587

**Published:** 2023-05-24

**Authors:** Owen Jones-Salkey, Zoe Chu, Andrew Ingram, Christopher R. K. Windows-Yule

**Affiliations:** 1Oral Product Development, Pharmaceutical Technology & Development, Operations, AstraZeneca, Macclesfield SK10 2NA, UK; oxj807@student.bham.ac.uk; 2School of Chemical Engineering, University of Birmingham, Birmingham B15 2TT, UK

**Keywords:** continuous direct compression (CDC), powder cohesion, loss in weight (LIW) feeding, blending, tabletting, continuous manufacture

## Abstract

The pharmaceutical industry is undergoing a paradigm shift towards continuous processing from batch, where continuous direct compression (CDC) is considered to offer the most straightforward implementation amongst powder processes due to the relatively low number of unit operations or handling steps. Due to the nature of continuous processing, the bulk properties of the formulation will require sufficient flowability and tabletability in order to be processed and transported effectively to and from each unit operation. Powder cohesion presents one of the greatest obstacles to the CDC process as it inhibits powder flow. As a result, there have been many studies investigating potential manners in which to overcome the effects of cohesion with, to date, little consideration of how these controls may affect downstream unit operations. The aim of this literature review is to explore and consolidate this literature, considering the impact of powder cohesion and cohesion control measures on the three-unit operations of the CDC process (feeding, mixing, and tabletting). This review will also cover the consequences of implementing such control measures whilst highlighting subject matter which could be of value for future research to better understand how to manage cohesive powders for CDC manufacture.

## 1. Introduction

The pharmaceutical industry is currently undergoing a paradigm shift from batch to continuous processing in order to improve manufacturing efficiency [1]. Continuous direct compression (CDC) (Figure 1) is considered highly efficient because of the reduced number of unit operations involved [2,3]; however, there currently exists only a limited range of formulations for which it is viable. Suitable formulations will generally have low drug load, a need for adequate flowability (as the material has to be transported through the system), and have sufficient tabletability, for the CDC process to work [4].

There are several critical material attributes (CMAs) which impact the flow of pharmaceutical powders and hence the successful formulation of oral solid dosage forms, of which cohesion is thought to be the most significant [5]. Powder cohesion refers to the affinity of particles to adhere to each other. It manifests in the bulk as resistance to powder flow often accompanied by adhesive behaviour, which refers to the tendency of a material to bind to surfaces and other materials [6]. Factors which contribute to cohesion include:Liquid bridges (liquid bridges will not be discussed in this literature review as there already exist comprehensive reviews of this topic [7,8,9]). Our focus in the present work will also be on the *effect* of cohesion in CDC processes rather than the *causes* of cohesion) [10];Van der Waals forces [11];Electrostatic forces [12];Frictional charging during handling [13];Particle shape and size [3].

Although humidity is controlled in a GMP environment, moisture is still present in the air and can be adsorbed and or absorbed into the powders, increasing cohesivity [14,15,16]. This increase in local moisture content in powders results in transient, complex interactions between powder surfaces and water, resulting in higher cohesion and decreased flowability [14,15,16]. Therefore, for every instance where good flowability is desired, an increase in humidity will have a negative effect. On the other hand, with compaction, there is evidence that some moisture can produce higher-strength tablets. However, too much moisture impacts the tablet strength negatively [17]. Thus, the positive effect of humidity, during tabletting, is largely outweighed by its negative effect across all the other unit operations in the CDC process. Therefore, generally or where suitable, moisture content should be kept to a minimum. In the interest of the review, relative humidity will be discussed when it is the only variable suited to solving an issue, for instance mitigating the triboelectric effect during powder feeding (Section 2). Furthermore, there are additional reviews [8,9] which discuss the influence of humidity on powder cohesion.

Active pharmaceutical ingredients (APIs) are typically the components of tablet formulations that give rise to handling issues, due to their often very high cohesivity, low bulk density, and highly aspherical geometries, in turn resulting in undesirable operating performance for CDC [6,18]. Many APIs are needle-like and very fine (<10 μm) resulting in a high contact area and cohesion (and thus poor flowability), whereas excipients are typically coarser and have a more favourable (spherical) particle shape, leading to better flowability. This difference in flow properties is perhaps unsurprising, as excipients are inactive substances whose purpose is to aid the formulation process by supporting or enhancing the stability and tabletability [19] of a pharmaceutical dosage form; as such, cohesive excipients would be omitted during the experimental design stages [20] as they are much easier to replace than APIs. Based on the above, it is clear that a high concentration of API may heavily affect the processability of powders, rendering the CDC manufacturing route non-viable [21].

In spite of the above, cohesion is not always a negative influence on CDC operations. In some instances, moderate cohesion has been found to aid the proper functioning of a given unit operation. For example, during powder feeding the addition of nano-sized silica was shown to result in reduced adhesion to feeder surfaces and improved powder flow as will be discussed later in this review [6,22]. However, in most cases of high cohesivity, flow is inhibited [23,24], which is problematic for feeding [25], mixing [26] and die filling [27,28]. Conversely, during the tabletting stage, powder cohesivity (specifically interparticle adhesion) is generally desirable, inviting smaller particle sizes and higher surface energy powders [29,30,31,32,33,34,35,36].

There are three main ways in which cohesion can be managed in tablet production:1.Formulation modification, through the introduction of glidants or lubricants [37].2.Operational process changes which are documented in many different studies for each unit operation [38].3.Granulation, which also manages cohesion as poor flowing powder fines are made into coarser more uniform agglomerates. (This is beyond the scope of this literature review as this unit operation is not a part of CDC processes [39]).

However, most literature does not consider how cohesion controls impact downstream unit operations. As far as the authors are aware there are no literature reviews that encompass the mitigation of the effects of cohesion and how those controls may affect certain unit operations. Therefore, the primary aim of this literature review is to explore and consolidate current knowledge concerning the impact of powder cohesion and cohesion control measures on all unit operations in the CDC process (feeding, mixing and compression), discuss the consequences of cohesion control measures implemented, and establish what further studies could be valuably conducted in the future.

## 2. Powder Feeding

### 2.1. Feeding Introduction

Powder feeding for CDC processes needs to be accurate and consistent for CDC performance success. Feeding modes are commonly defined by the delivery and maintenance of material, in a given time frame. As such, there are two main categories for feeders: volumetric and gravimetric, which supply material according to volume per unit time, and mass per unit time, respectively [2,22,25,40,41]. Volumetric feeding is often used to understand the characteristics of powder behaviour within the hopper, whereas gravimetric feeding seeks to smooth out any volumetric perturbation through feedback control, arguably resulting in gravimetric being the superior of the two modes for mass delivery [2,22,25,40,41,42]. Powder feeding is the first unit of operation in the CDC process; thus disturbance or variation from the set point can propagate downstream, fundamentally impacting the tablet quality [25,43].

One of the tasks of the blending stage in ensuring that constituents are well-mixed is to smooth out the perturbations from the feeder. The lower the intensity and frequency of these perturbations, the lower the chance that content uniformity or manufacturability issues will arise downstream [22,44,45]. Moreover, it should be noted that not all disturbances result in a detrimental impact. It is both dependant on the magnitude and duration of such disturbances—Gyürkés et al. [46] aptly discuss this concept with the use of funnel plots.

Powder cohesion impacts feeding performance by introducing variation through complex bulk behaviour, which results in the variation of flow behaviour [22,25]. This contributes to both the restriction of power into the converting screws, and the resistance to flow within the swept volume of the screws. In addition, these materials are prone to electrostatic charging, causing attraction-repellent behaviour, which can lead to deposits of material residing and adhering to surfaces post-feeding [15,47]. This buildup can migrate into the bulk feeder, dosing it with a spike of material, therefore resulting in a compositional imbalance [15,47]. The following section aims to discuss the work of several publications, exploring the complexity surrounding powder feeding.

### 2.2. Feeding Discussion

The difficulty associated with powder feeding directly derives from the handling of larger volumes of powder (in the hopper) and converting this into smaller, consistent streams (at the outlet). Essentially, this refers to the driving of powder down a pathway of lowering power-to-powder contact, to instead increase powder-to-wall contact. When there is resistance to this gradient of increasing powder-to-wall interactions, it suggests there is a lack of flowability, which commonly manifests in the form of cohesivity. Though cohesion is ultimately a microscopic process (i.e., it occurs at the scale of particle-particle contacts), at present its effects are predominantly characterised via macroscopic (bulk) measurements [48].

The Flow Function Coefficient (FFC) is a measurement used to quantify the amount of stress required to cause a powder to yield, and is one of several manners in which a powder’s ‘flowability’ may be characterised [49,50]. Gathered from a ring shear tester, for example, the measurement applies a normal consolidation force to a powder bed and measures the tangential force required to cause failure (incipient flow). A yield locus of shear stress vs normal stress is plotted and, applying Mohr–Coulomb failure theory, the major principal yield stresses of consolidated and unconfined material are obtained. FFC is the ratio of these values. The theory and methodology is discussed in various degrees of detail in the following papers: [5,49,50,51]. Leung et al. [5] performed a comprehensive analysis on 1130 ring shear measurements and effectively surmised that particle friction coefficients are negligible on FFC measurements, suggesting that cohesion is the primary characteristic which influences the measurement. In addition, the authors state that mitigating the effects of cohesion is far more influential than mitigating the frictional aspect(s) of the powder when it comes to improving powder flow, and also suggest that altering interparticle forces is the best approach to take when combating cohesive powders.

Leung et al. [5] demonstrated the effect of introducing 1% *w*/*w* of colloidal silicon dioxide into two excipients on FFC, cohesion and angle of internal friction. The first excipient—hypromellose—showed a significant drop in cohesion and an increase in FFC (Figure 2). Meanwhile, the second excipient—dicalcium phosphate anhydrous—showed the inverse behaviour, but to a much lower degree. No significant changes were seen in the angle of internal friction for both materials. Additional work regarding the use of colloidal silicon dioxide nanoparticles for formulation adjustment is seen in [52,53]. The mechanism at work involves the surface coating of SiO${}_{2}$ nanoparticles, which introduce additional distance, therefore lowering the strength of the van der Waals attraction between the two surfaces of the more cohesive species [5,6,52,53]. Furthermore, Leung et al. state that particles with uneven shapes and rough surfaces may receive little beneficial effect from the SiO${}_{2}$ surface coating, and therefore minimal improvement in flowability. The SiO${}_{2}$ will fail to act as a barrier to the contact of the carrier species as it is shown that using SiO${}_{2}$ as an additive can be detrimental to FFC.

Tran et al. [54], investigated these underpinning mechanisms which govern flow modification through the use of colloidal silica—their findings agree with those of Leung et al. [5]. Tran et al. studied the differences in silica loading for two different excipients using flow characterisation, static image analysis, and particle size distribution. It was found that materials with higher surface area (described as rugged) were able to accommodate higher silica loads, and that the optimal amount of silica loading is dependent on both the surface area and the size of the carrier particle but, most notably, their results discussed the possibility of taking a ‘quality by design’ approach by determining silicate (glidant) loading by carefully assessing the carrier powder’s physical properties.

Lopez et al. [55] published a DEM study on the effect of increasing cohesion on feeding performance. The researchers began by calibrating the powder used in the system, first identifying and assigning the physical properties and particle shape of paracetamol, before performing a sensitivity analysis across different adhesive stiffness values ($K_{adh}$) and conveying impeller rotation rates. When the conveying impeller was kept at a constant rotation rate (10 rad/s), the particles saw an increase in translational velocity in conveying barrel with increasing $K_{adh}$ (ranging from 0 to 0.5). This effect was lesser but present on the interface between the conveying barrel and hopper. On the other hand, keeping $K_{adh}$ at a constant 0.2, across increasing conveying speeds (10, 20, and 50 rad/s) saw an increase in the overall translational velocity of the particles. Perhaps unsurprisingly, for the range of parameters explored, increasing conveying speed was observed to have a greater influence on particle velocity in the hopper than altering cohesion.

The DEM simulation of Lopez et al. [55] also provides valuable insight into the expected power draw; given the CDC system is continuous, it would mean the process would be running for days, if not weeks, at a time, making this an important consideration. The authors demonstrated the average mass discharged per Watt as a function of the $K_{adh}$ (between 0.1 to 0.5 $K_{adh}$), which showed an exponential decay with increasing $K_{adh}$ Figure 3. Accordingly, $K_{adh}$ values higher than 0.5 showed arching: a phenomenon which occurs when the cohesive forces between particles are stronger than forces due to gravity. Arching (also referred to as bridging) is a semi-stable instance of particles forming particle–particle structures suspended in a hopper or opening, which are resistant to the displacement of powder beneath them [25]. For additional information on the influence of particle properties on arching, see the following papers: [25,49,56]. Lopez et al. described this behaviour with absolute translational velocity where they revealed instances which lead to inconsistent feeding of the conveying screw, which in turn lead to flow irregularities Figure 3. The authors link this artefact of transient macro behaviour to the micro by stating that inconsistencies in flow result in the improper filling of the screw pitch, giving rise to an inconsistent mass flow rate. This is best demonstrated by the plot of mass per screw pitch vs. $K_{adh}$ shown in Figure 3). Finally, the researchers indicated what future work would be of value, notably suggesting the mapping of the transient cohesive behaviour by altering the particle properties, feeder’s geometry and rotational speed.

To build upon the above suggestions, it would be interesting to see the implementation of a DEM study on a more commercially representative feeder, which possesses both a hopper agitator and twin screw conveying elements. Allowing the exploration of concepts such as agitator design and rotation speed to increase barrel/screw filling consistency or the breakage/mitigation of arching. What is more, the research conducted by Leung et al. [5], which discussed the minimal effect of friction in powder flow consistency, could utilise simulation tools such as DEM to more directly support their claims.

Escotet-Espinoza et al. [6] present a publication showcasing the use of silication to improve feeder performance. The researchers began by pre-blending the three different APIs explored with 1% silica, before characterising the bulk behaviour of the pre-and post-silicated API. The APIs (and their silicated counterparts) were then fed through the feeder, whilst a catch scale continuously measured the mass exiting the feeder. The response was then measured over time, with attention being paid to the consistency and accuracy of the feeding process.

Like Leung et al. [5], Escotet-Espinoza et al. [6] also found that the introduction of silica improved flowability by reducing the interparticle cohesion. Feeding improved with the addition of silica, with each of the API case studies showing improved screw speed consistency, a reduction of mass flow RSD (relative standard deviation), a reduction in powder adhesion to the hopper surfaces, and a reduction of remaining mass. This suggests that the addition of silica is hugely advantageous for feeding. The improvement can potentially be attributed to an increase in flowability, allowing the powder to better fill the swept volume of the screws, thus leading to an increase in mass per revolution with the addition of silica. This observation is supported by the work of Lopez et al. [55] which shows that, with a reduction of $K_{adh}$, there follows an increase in mass per screw pitch. Furthermore, with FFC (as previously discussed) being a strong indicator for powder cohesion, it would suggest that it would be useful to understand this screw-filling behaviour. This conveniently leads to another suggestion to further Lopez et al’s work [55], whereby similar DEM studies could be undertaken to understand the effect of additives on feeding and powder conveying. Escotet-Espinoza et al. [6] also describe the material used generating electrostatic charge, which enabled the API to stick to the agitator. However, in the silicated API case study, there was very low powder adhesion, suggesting that adding silica also suppressed the effects of electrostatic adhesion.

A study by Lumay et al. [57] explicitly investigated the influence and mechanisms of mesoporous silica (MPS) on the electrostatic charge, again finding that the addition of silica species improves powder flow. The experiments differed from those discussed previously in several manners. Firstly, the study differed in the materials and methods used—the authors tested three different grades of silica, varying in particle and pore size, in three common excipients (microcrystalline cellulose, lactose, and maize starch) using a rotating drum. Secondly, the authors applied different experimental analyses: the dynamic effect of silication was evaluated in this case by measuring the cohesive index [58] of materials (with and without MPS) across a range of rotation speeds. Finally, through the use of a standardised charge density measurement technique [59], the charge density of the material blends is shown in the presence of different types and amounts of MPS.

It was found that the MPS with the smallest particle size provided the greatest improvement in flowability. Furthermore, the addition of just 0.5% *w*/*w* made a considerable impact on the cohesive index—see Figure 4A. The authors attribute this improvement to the particle size of the MPS, explaining that there would be an increase in effective surface area, which would suggest greater coverage, and thus intervention between, cohesive species.

An additional striking finding of the study was that the addition of silica was also found to change the shear-dependent rheological properties of certain tested materials, with typically rheopectic materials becoming thixotropic when mixed with a small volume of silica. Specifically, the study evaluated the cohesive index of the powder dynamically by testing the excipients against their 2% *w*/*w* silica counterparts across a range of rotation rates (see Figure 4B). Both materials, maize and lactose, were shown to be more cohesive (thickening) with increasing rotating speed (ranging from 2 rpm to 10 rpm) and in the presence of silica, less cohesive (thinning) with increasing rotating speed. The silicated maize showed a greater decrease in cohesion than the silicated lactose with increasing rotation speed.

Finally, the inclusion of MPS was shown to decrease the static charge density of the powder—see Figure 4C. Across all materials and MPS grades, the addition of 1% (of MPS) saw similar results to the addition of 2—suggesting 1% was sufficient for charge mitigation, with any greater amount being surplus. MPS incorporated into maize saw a decrease in charge after just 0.5% *w*/*w*, with the grade of MPS used making a minimal difference. Conversely, MPS in lactose still produced a decrease in the magnitude of charge density, though different grades responded differently. This suggests there may be some complexity associated with the use of MPS with lactose. It is potentially important to note that Lumay et al. [57] stated that the lactose varied in positive or negative charge on the day, which was dependent on the air’s relative humidity. Given the magnitude is roughly the same but the charge differs, if the S244 (Syloid^®^FP244) was positive it would suggest that the grade of MPS used would have little effect on the charge density and vice versa for SXDP (Syloid^®^XDP3050). Research by Ramires-Dorronsoro et al. [60] supports this: when completing a similar study—with a different measurement technique—they also witnessed spray-dried lactose exhibiting a positive charge density. Ultimately, this suggests that the use of silica in poorly flowing hygroscopic powders greatly improves the flowability, and decreases the sensitivity to triboelectric charging.

The previously discussed papers showcase the underpinning behaviour of cohesive powders during feeding, and the discussion of silication demonstrates the advantage of understanding the mechanism underlying a powder’s cohesive properties. Despite this, when considering CDC in its entirety, the predictive capability is a large part of reducing experimentation and understanding the process. Therefore it is desirable to be able to predict the feeding performance from a few key powder characteristics [42,43,61], one of which is cohesion.

Van Snick et al. [43] and Garcia-Munoz et al. [45] provided some of the first in-depth analysis of the end-to-end CDC process, thus by definition including aspects of feeding. The authors discuss the correlation of bulk density, tapped bulk density and FFC to feeding consistency (Figure 5). The experiments involved the comparison of different materials running volumetrically (at a constant screw speed) from full to empty, whilst measuring the feed factor response over time. The maximum ($ff_{max}$) and minimum ($ff_{min}$) feed factors (g/revolution) were then taken and used for further calculation. Feed factors change throughout the hopper fill as powder above the hopper bears down upon the powder below, due to gravity. When the hopper is at low fill, there is not only a reduction of gravitational bulk powder compaction but a potentially inconsistent feed into the conveying volume.

As a result, bulk density increased linearly with max feed factor, tapped bulk density increased linearly with delta feed factor (where delta feed factor is: $ff_{\Delta }=ff_{max}-ff_{min}$), and finally flow rate variability exponentially decayed with increasing FFC. Since both bulk density and FFC are common descriptors for powder cohesion, it then can be used as a proxy to describe behaviour. The study highlighted FFC values greater than 3 would give sub 1% flow rate variability, and that powders with low bulk density and low tapped bulk density are prone to large $ff_{\Delta }$—which suggests instability when processing. Escotet-Espinosa [22,62] also looked into both the influence of powder properties on feed factor and modelling such behaviour using the following (simplified) equation:(1)$$ff_{w}=ff_{max}*\left(ff_{max}-ff_{min}\right)exp\left[-\beta *w\right]$$
where $ff_{w}$ is the feed factor as a function of ${}^{\prime }w^{\prime }$ the weight, while $ff_{max}$, $ff_{min}$ and β are the maximum feed factor, minimum feed factor, and a fitting constant based on feeder geometry, respectively.

The $ff_{max}$ and $ff_{min}$ were then plotted in a correlation heat map (Figure 6), relating the responses to a plethora of recorded bulk characteristics. Interestingly, the highest correlation was seen for cBD, conditioned bulk density (g/mL). cBD is akin to tapped bulk density, obtained from the initial standardised conditioning step of a powder rheometer. It involves a blade passing through a volume of powder with a set torque before measuring the weight of the known volume. Paying attention to the units, it is a given that there is a correlation between the two density measurements, but the interest is with ‘compressibility at 15 kPa’ which also scored high for $ff_{min}$, which would be a method for attaining higher bulk density. This suggests that powder that is capable of being compressed, and has a high conditioned bulk density, will deliver a high $ff_{min}$. Similar studies by Shier et al. [63] and Bekaert et al. [42] similarly look to predict feeder performance using bulk characteristic properties.

Engisch & Muzzio [41] performed a study on varying hopper refilling conditions on powders with varying cohesiveness and measured the resultant effect on feeding accuracy and consistency. Typical Loss in Weight (LIW) feeder operation is gravimetric, until topping up (or refilling), where the system operates volumetrically to ensure a consistent feed is maintained whilst an influx of mass is registered on the feeder’s load cell. This method results in a trade-off between the number of times the system has to enter the ‘less accurate’ volumetric mode and ensuring the hopper is sufficiently full, such that the feed factor is stable. For an example of the effect of decreasing fill on feed factor, see Figure 5. The conclusion from Engisch & Muzzio is that refilling the feeders with less material, and therefore more frequently, results in better overall performance. This should be kept in mind when paying attention to the duration of time that the system enters volumetric mode and reducing the overall disturbance from the setpoint over the whole duration of running the feeder. Irrespective of the frequency and volume of fill, Engisch & Muzzio state refilling should be gentle as this causes the least disruption to feeding. Furthermore, it is shown that refilling sensitivity is much lower at higher refill levels, Figure 7. Refilling at higher fill levels reduces the maximum set point deviation, time of deviation, and the total amount of mass during deviation.

A similar methodology could also be applied to combat cohesive powders through compaction during refilling, by attempting to keep the hopper volume as constant as possible, during their most stable $ff_{w}$. To build a theoretical case study, using the APAP being fed volumetrically in Van Snick et al. [43] (Figure 5) and the supporting evidence of refilling stability from Engisch & Muzzio [41], see Figure 7. A top-up of 25–30% of the volume of the hopper could be used and triggered at the 50–60% fill level, refilling to 75–80%. This would occur during the most stable feed factor window, whilst the hopper is already half full. During the window in which powder is dumped into the system, there is a transient influx of powder onto the existing powder bed. This impulse of vertical pressure on the powder would increase powder compression within the bed, increasing the bulk density of the powder local to the conveying screws. Corroboration is seen by adding Escotet-Espinosa’s [22] correlation measures, as there is an improvement in the conditioned bulk density, which increases $ff_{min}$ reducing the $ff_{\Delta }$, essentially reducing the feeder inconsistency related to lower fill levels. Albeit this theoretical case study is generalising by not considering phenomena such as disturbances due to air flows during discharging mass, and an increased volume of powder that may be prone to triboelectric charging due to the agitator. DEM studies, such as Lopez et al. [55], open a stage for exploring some of the fundamental behaviours seen in these systems, in particular when looking to evaluate the micro and macro behaviour attributed to conveying inconsistency.

Finally, Yadav et al. conducted a comprehensive study with a loss in weight feeder [64] utilising a range of screw speeds, gear ratios, and screw types, across a range of materials. The result was a PCA model which identified the influence of these processing parameters and how they interacted with the materials’ bulk characteristics. The principal components consisted of one comprising the powder’s bulk characteristics, a second representing the feeder’s processing parameters, and a third representing an interaction component. Most notably, the interaction component related screw free volume (${\mathrm{cm}}^{3}$) to density, particle size ($d_{10}$, $d_{50}$, $d_{90}$), wall friction angle, and Feed Factor. The authors note that a different screw type will have a different screw-free volume, meaning that the comparison between a smaller or larger volume to dispense per revolution must be taken into consideration when evaluating the feed factor. Similar findings are seen with Engisch et al. [2], where their suggestion is the use of both self-cleaning concave screws and the removal of any outlet screens for very cohesive materials.

### 2.3. Feeding Summary

Cohesion has been shown, across multiple studies, to worsen the flowability of powders, in turn reducing the consistency with which powder flows into the conveying volume [5,43,55]. The transient inconsistency of material flowing into this volume persists and has been shown to result in turn in an inconsistency in the delivered mass powder flow—and therefore a deviation from the desired set point [2,41,55,65]. Thus, it is very important to ensure that powder, despite its cohesive nature, is consistently motivated to enter and be transported through the conveying section of the feeder.

This review outlines just a few examples that have been used so far to reduce the negative influence of cohesion on feeding: reducing interparticle contact between cohesive species [5,57]; reducing the effect of triboelectric charging [57]; refilling the hopper at moderate fill levels [41,65]; and adjusting screw speed and screw type [2,64].

When applying the knowledge gained from Lumay et al. [57] to the work of Escotet-Espinoza et al. [6] and Lopez et al. [55] it would perhaps be sensible to suggest that smaller sized silica nanoparticles would be ideal for improving flowability at this stage of the CDC process. Silica nanoparticles have also been shown to change the blend from shear-thickening to shear-thinning, implying that, when the powder contacts the hopper’s agitator or the conveying screws, one might expect a further transient, localised increase in flowability.

Despite silicates holding the much of spotlight for discussion, surrounding improving powder flowability, it should be noted that several alternative materials could be selected. However, every material in question, for flow modification, would need to undergo consideration for its end-to-end impact on the process. Silicates, for instance, have been shown to negatively impact compatibility [66,67] but also improve flowability and therefore improve feeder processing consistency [5,43,55], whereas magnesium stearate has also been shown to increase flowability, but due to its hydrophobicity—coating the surface of the API—would hinder the dissolution of the API within the formulation [68]. This would result in the material being greatly undesirable for any scenario which may involve the intimate blending, or over-mixing, of magnesium stearate with API. However, pre-blending API with other materials such as tricalcium phosphates or microcrystalline cellulose [69,70] propose the ability to improve flow behaviour at a minimised trade-off, and as such, more research should be carried out into both the use (for improving feeding performance) and the end-to-end implications of selecting such materials.

Our review of the literature has also highlighted the benefit of future DEM studies which could be undertaken to better understand the fundamental powder movement that governs powder feeding, in particular using commercially-relevant geometries. Notable areas of study could include modelling the effect of silication; altering, and potentially optimising, agitator and screw design; further investigation of the influence of hopper refill regimes. The development of dynamic DEM simulations could also be used to develop and test various control strategies, both existing and novel. Intuitively, it could be possible to develop a control proposition for shear thickening powders involving a sudden reduction in RPM followed by a slow build-up of screw speed back to setpoint.

## 3. Blending

### 3.1. Blending Introduction

Despite diverse designs available commercially and in the academic literature, continuous blenders can, in general, be reduced to four main components: an inlet, an outlet, a mixing volume, and a mechanism for promoting blending whilst in the volume of the blender. Typically, this last component will be some form of rotating agitator, the geometry of which may differ from mixer to mixer, or even formulation to formulation. Recognising the different types and geometries of mixers that exist within this space is fundamental when evaluating the role of powder mixing in CDC performance, as the features that differentiate one mixer from another can have significant impacts on the behaviour of the powders being blended, and thus may require different measures to handle the blending of cohesive powders.

Achieving good mixing is integral to the performance of any given pharmaceutical tablet; the therapeutic efficacy and mechanical properties of the tablet will suffer without the effective distribution of its constituent materials, ultimately compromising the safety of the product [43,71,72,73,74,75]. Thus, mixing inherently defines the final quality of the formulation (albeit that the final product cannot be delivered without the consistency and accuracy of the upstream and downstream processes [43,45]).

Practically defining whether a formulation is well-mixed is a complex endeavour with issues such as frequency of sampling [71,76,77], location of sampling [71,76,77], and scrutiny (size) of the sample [71,76,77]. However, describing the mechanisms of mixing may be ever-so-slightly easier, and is of more direct relevance to the present work.

The two commonly-defined types of mixing—macro- and micro-mixing—are both essential when looking to create ideal mixtures [71,72,75,78,79]. Macro-mixing is responsible for large, global movements, of material—or bulk transport—allowing the material to flow and circulate around the vessel [71,75,79,80,81,82]. Micro-mixing, on the other hand, involves the movement and displacement of material at the particle scale, i.e., the local interactions between particle species [71,79]. Accordingly, micro-mixing achieves the detail of the mixing action due to the mixing happening at the smallest possible scale: the particle–particle level [71,79], whereas macro-mixing serves to smooth out the upstream mass flow rate perturbations, due to the flow and bulk motion of the powder [71,79].

Fan et al. [83], succinctly discuss the evaluation of mixing in terms of cohesive mixtures. Figure 8 and Equation (Equation 2) encapsulate the majority of this discussion. It is shown that by increasing the surface force—which is also a function of particle size, and therefore surface area—it is possible to reduce the influence factor defined in Equation (Equation 2). A low influence factor means surface forces dominate, and the particle species adhere to the surface of another particle species, creating a perfectly-ordered blend. Small, less dense particles would be the ones to adhere to the more dense ‘carrier’ particles- due to gravitational forces. Then, if the surface forces are minimal, gravity dominates, and thus the mixture would involve non-cohesive species, following more typical solids mixing criteria.
(2)$$Influence\;Factor=\frac{Gravitational\;Force}{Surface\;Force}$$

Conversely, de-mixing occurs through several mechanisms, such as: segregation, the separation of a formulation into constituent particles of similar properties, for example, size and density [71,76,84]; agglomeration, the formation of particle–particle dense macro-particles, due to interparticle attractive forces [85,86]; and electrostatic repulsion, or attraction, due to inter-particle charge imbalances, which predominantly result from contact-induced electron transfer, as particles move relative to each other and the equipment surfaces [26,87]. The part cohesion plays differs significantly in each of these mechanisms, and can either subdue or exacerbate de-mixing [84]. For instance, electrostatic charges could be worked into powder, through the triboelectric effect, creating attractive or repulsive forces between different species [26,77]. These interactions underpin much of the following discussion of how cohesion affects the process of mixing and the competition between mixing and de-mixing.

### 3.2. Blending Discussion

Processing parameters such as agitator RPM, feed rate (throughput), and agitator design, are the primary methods of manipulating powder through a given blending volume [43,75,88]. For example, trends have been identified for improving mixing quality through the use of processing parameters: macro mixing as a function of mean residence time; micro mixing as a function of RPM (agitation); residence time being a function of throughput [43,75,82,89,90]. These examples signify the importance of process parameters on mixing performance. However, the intricacies of powder behaviour in response to these systems are seldom discussed in the wider literature.

As discussed in the preceding section, cohesion can be overcome through the use of formulation additives and/or changes to the process [5,37,57,78]. Additives, such as glidants (e.g., SiO${}_{2}$), are typically used to make the bulk powder properties easier to deal with, by increasing flowability [5,37,57]. However, additives change the overall formulation composition, which can have detrimental effects on downstream processing-namely tabletting [66,67]. In contrast, process changes (processing parameters) can potentially overcome issues relating to cohesion without altering the formulation’s composition [43,75,88]. Utilising these favourable processing parameters is thus the more practical of the two solutions for CDC processes, but the development and transfer of knowledge between different systems appears to be a substantial hurdle. This is largely due to this approach requiring a much deeper understanding of the specific unit operation being used in the study. Therefore, the knowledge gained is often less transferable; each particular set of process parameters may be unique to system geometry, orientation, or method of operation. With this in mind, it is useful to note the use of dimensionless numbers/groups within the discussed research, as it allows greater understanding and application of relative powder behaviour between publications.

Portillo et al. [91] performed trials on an axial continuous blender (see Figure 9)), the axis of which was adjustable between ±30${}^{\circ }$ to the horizontal. The researchers found that at (+30${}^{\circ }$) incline there was an increase in mean residence time which led to a decrease in the relative standard deviation (RSD) of Acetaminophen (APAP) concentration in two grades of lactose. This demonstrates the effect of longer mean residence time on macro mixing and a reduction in APAP RSD. The grades of lactose differed in terms of average particle size and size range: Lactose 100 (70–250 μm, average 130 μm) and Lactose 125 (55 μm), whereas APAP (36 μm) remained the same in both tests, and made up 3% *w*/*w* of the mass flow rate. They also discussed that, despite typical behaviour, the smaller (and more cohesive) grade of lactose did not affect mixing performance at either the high or the low rotation speeds. It was observed that while agglomerates were readily formed by the cohesive material, these were relatively weak and agitation from the impeller was sufficient in breaking the clusters up, dispersing the powder.

Later work by the same group [78] discussed the influence of cohesivity on powder behaviour in a different continuous mixer (albeit similar to the blender shown in Figure 9), inclined at +17°. The study used two different grades of lactose, Fast Flo and Edible, which displayed flow indices (measured using a Gravitational Displacement Rheometer) of 24.9 and 34.8 respectively, suggesting them both to be highly flowable powders. In addition, the bulk density tapped density of the Fast Flo Lactose were 0.626 (g/mL) and 0.704 (g/mL), respectively, while the Edible Lactose measured 0.629 (g/mL) and 0.981 (g/mL), respectively. Portillo et al. [78] found that cohesivity negatively impacted axial mixing, becoming a statistically significant factor at higher levels of agitation; thus demonstrating cohesion’s effect in resisting the separation or relative movement of particles. In addition, the authors observed that particles with high cohesion experience longer residence times, with similar path lengths, suggesting the particles move at a slower velocity through the blender. This postulates that methods which increase the shear rate, and therefore the relative particle velocity, may combat the effects of cohesion by increasing dispersion [92], and consequently improve micro-mixing. This could also be described as increasing the dynamic granular temperature as a function of the increased shear rate [93,94]. This finding may, at face value, seem to contradict the findings discussed in the previously mentioned study by Lopez et al. [55], in which the translational velocity of the powder going through the conveying portion of the system was observed to be higher with increasing cohesivity. However, despite the *net* velocity being higher, the *relative* velocity (which drives micro-mixing) may nonetheless be lower.

It should be noted that any residence time distribution (RTD) data will be a function of both the volume % of the tracer and the interaction of the tracer within the bulk that it aims to measure. For more detail on RTD behaviour in continuous blenders see Escotet-Espinoza et al. [61,80], where the authors provide a comprehensive two-part study, analysing the behaviour of tracers with different properties.

Vanarase et al. [89] investigated the blending performance of a horizontal linear blender, both with and without the addition of a co-mill. The authors stated that under optimal macro-mixing conditions, micro-mixing was identified as the limiting factor, leading to poor mixing performance. The inverse was also found for optimum micro-mixing conditions. This highlights the independence of these two types of mixing, and the consequent requirement, to balance a blending system’s capability to deliver both sufficient micro and macro mixing.

Interestingly, research by Portillo et al. [91] contains contradictory findings, when compared to Vanarase et al.’s study [89]. They found that using an incline blender increases the overall residence time when compared to a horizontal system—see Figure 10. Thus, for the same residence time as was demonstrated by Vanarase et al. and therefore similar macro behaviour, Portillo et al. showed that the agitator speed could be increased, which in turn provides greater shear, achieving improved micro-mixing conditions with cohesive powders.

Van Snick et al. [43] investigated the performance of different mixing blade configurations (Figure 11, graphically demonstrates the orientation of blades along the agitator’s axis) and found that despite the ‘P16’ blade configuration not delivering the highest dispersion, it offered the best consistency (see Figure 12). The ‘P16’ blade config’ utilised a series of two sets of transport blades (45°), followed by a set of radial mixing blades (90°). Whereas the other example—‘D8’—delivered the highest powder dispersion, with the downside of higher variability. This is supported by Vanarase et al.’s [89] findings, with both surmising that a balance is required between micro and macro mixing. Based on ‘P16’s performance, in Van Snick et al. [43], it seems as if the ‘D8’ configuration provides a surplus of dispersion-lending to poor macro conditions. The authors also attribute the good performance of ‘P16’ to the increased mass hold-up within the blender’s volume, as the mixing blades are shown to increase fill level, and therefore total mass hold-up [43]. Ultimately, the study suggests that varying position, number, and angle of the blades could aid in controlling API within the blending phase, and that different blade configurations may be selected according to the formulations’ properties, such that the desired throughput is attained, whilst output variation is minimised.

Gao et al. [95] developed a two-blade (periodic) section of a continuous horizontal mixer (see Figure 13), and evaluated mixing performance using a monodisperse DEM simulation of the blender. Gao et al. [95] subjected four different simulated powders (classified as: control, lower density, larger particle size, and cohesive) to the same tests, and mapped the axial conveying efficiency in fill level-rpm space. This conveying efficiency was quantified by normalising the mean axial velocity of particles with the rotor speed and effectively giving the mean distance conveyed per revolution. The powders were subjected to various fill levels (from 25–75%) and a range of rpms (50–250). Figure 14 shows that the relationship between normalised axial speed, and fill level/rotational speed was very different for the cohesive simulation: axial velocity is lower, when compared to the control and to the other powder types, especially at low speeds (50–125 rpm) with a moderate-to-high fill level (50–75%). In addition, the cohesive particles achieved a max velocity of only 38 mm/revolution, the lowest of the four samples [95]. These results are in agreement with the discussion gained from the work of Portillo et al. [78] work.

It is worth noting that, for all non-cohesive powders used in Gao et al.’s work, when going from high to moderate fill levels (i.e., from 75% to 40%) at 100 rpm, normalised axial velocity increases. Cohesion is still shown to have an effect at higher rpms, demonstrated by the increased distance between the contours. This finding corroborates with the rpm-cohesion-significance result from Portillo et al. [78], clearly showing that the relative effect of cohesion will depend on operating conditions (speed and fill/mass rate), and not necessarily in a predictable way, reinforcing the need for more work in this area. This finding, surrounding the three non-cohesive powders, was further supported by the work of Sarkar and Wassgren [96], where the normalised axial velocity presented the same trend, despite different geometry, particle properties, and operating conditions.

Gao et al. [95] also mapped the continuous blending rate ($k_{c}$), which is the ratio of the normalised mean axial velocity to the time-dependent RSD decay rate, where $k_{c}$ essentially details the ratio between the axial (macro) and radial (micro) behaviour of the system. The results obtained were a similar contour plot to Figure 14, but in terms of $k_{c}$. Across the different particle properties, it was shown that high RPM and low fill produced the greatest $k_{c}$. Moreover, there was little difference seen in $k_{c}$ at <125 rpm irrespective of fill level. For rpm > 125: going from high-to-low fill showed a sluggish increase in $k_{c}$, as the space between contours and colour changes progressed slowly. This highlights that cohesion does not only affect the axial aspect of the mixing process, but also contributes to the hampering of radial mixing. Finally, Gao et al. [95] suggest further investigation on segregating materials, which would prompt several questions on a constituent’s experience within the bi-or-polydisperse system. Particularly, it is worth considering what might be the best excipients to pair with different levels of cohesive materials, in order to best aid their macro- and micro-mixing. This has the potential to result in a CDC formulation, of the same drug, differing from their original batch formulation. Different excipient(s) may offer a significantly improved CDC performance, without impacting the tablet’s therapeutic efficacy.

Tomita et al. [97] utilised an atypical blending set-up, operating with both an impeller and a scraper in a horizontal mixer (see Figure 15). The inclusion of the scraper allowed for the control of the mean residence time in a manner that—unlike the previously-discussed systems—was quasi-independent of the magnitude of the impeller’s agitation. In this system, the rpm of the axial impeller was driven separately to the ribbon-shaped scraper, which circumscribed the internal wall of the blender. Figure 15B shows the RTD curves of spiked acetaminophen using different scraper rpm speeds, whilst running at a feed rate of 10 kg/s and constant axial agitator speed of 3000 rpm. There is a considerable difference in the RTD performance between the slowest (5 rpm), and highest (50 rpm) scraper speed, which delivered a mean residence time of 84.3 s and 12.1 s, respectively. The mixer provides a method of achieving higher micro-mixing whilst managing the macro-mixing through either increasing or decreasing the mean residence time. A similar result could be achieved with both the horizontal, or incline, blender through the alteration of either the angle of inclination, and/or the quantity of blades, blade angle, and/or blade positions. These design choices are, however, less practical to implement within a feedback loop or control strategy—though the incline angle of a blender may conceivably be varied dynamically.

Despite the advantages of an incline blender from a high-shear perspective, practical considerations must be made when attempting to maximise mixing capabilities. For example, the use of two units in series (horizontal blender and co-mill), as shown in Vanarase et al. [89], would allow independent, de-coupled, control in real-time. This reduces the risk that the process would leave a state of control, as each unit would have the capacity to compensate for the other. Conversely, if the incline blender is the only unit responsible for delivering both micro and macro-mixing, given it has the capacity to do so, it is likely the domain of control will be smaller when compared to the two units in series. This is because the process parameters controlling the micro-mixing also control the macro-mixing, and so there is a smaller window for advanced process control to operate within to ensure the mixing balance is maintained. This, therefore, showcases the strength of Tomita et al’s [97] blending unit, as it possesses the ability to independently influence the macro and micro aspects of mixing, in real-time, using process control.

Palmer et al. [75] evaluated fill level and content uniformity across a range of processing parameters in an inclined (+15°) continuous blender. These processing parameters include: throughput, number of mixing blades, and agitator RPM. The study then explored the same parameters, using the same formulation, with different grades of API (APAP)—see Table 1. The result is the proposed exponential decay model (see Figure 16), and micro-mixing model (see Figure 17), which uses both the Peclet Number (Pe, see Equation (Equation 4)) and the strain (ϵ, see Equation (Equation 3)). In addition, Figure 18, showcases a series of contour plots describing the effect of processing parameters on the fill level, for each APAP formulation. The study explored a range of Froude numbers (Fr) between 1.5, and 13.4 (for 150 and 450 rpm, respectively).

Comparing the bulk characteristics in Table 1, Micronized APAP (mAPAP) has closer bulk characteristics to Powdered APAP (pAPAP), but shows a unique shape contour in Figure 18. Despite pAPAP and Special Granular APAP (sgAPAP) being opposed in flowability indices, they present similar contour shapes. This indicates a similar fill-level, and therefore bulk powder response, to the number of radial mixing blades (see Figure 11) across a range of RPMs. For the mAPAP, however, increasing the number of radial mixing blades has more impact on the fill level at lower RPMs, and this impact reduces with increasing RPM. Van Snick et al. [43], discussed the necessity of sufficient mass hold-up (fill level) to attain good levels of macro mixing. The combination of these two studies suggests (i) that it is critical to determine the fill level landscape for the desired formulation to achieve and maintain sufficient mixing, and (ii) that different highly cohesive powders would be expected to exhibit different complex behaviours. What is more, using the mAPAP contour from Palmer et al. [75], considerations have to be made regarding what variables can be controlled in real-time. Since the radial mixing blades cannot be changed in real-time, it could be inferred from Figure 18, that using the highest number of mixing blades would be more optimum, as it would allow the system to have the largest range of fill levels.

Furthermore, Figure 16, from Palmer et al. [75], models the exponential decay of content uniformity RSD with increasing Strain (Strain, Equation (Equation 3)). Strain is a confounded variable for this system (an incline (+15°) linear blender), as increasing the RPM decreases the mean residence time, meaning that the number of blade passes also affects the mean residence time [75,91]. Thus, strain is dependent on the throughput, RPM, and the powder’s material properties, making it essential to understand how the powder responds to processing parameters. Understanding in this area can be gleaned from the work of Portillo et al. [78,91], Gao et al. [90], and Lopez et al. [55] regarding powder velocity, fill level, and resultant changes in residence time due to cohesion. More cohesive powders would be subject to longer residence times, but lower relative velocities. Therefore, more cohesive powders should be more resistant to increasing RPM in turn reducing cohesion’s effectiveness; Figure 18 demonstrates the added complexity. Ultimately, strain is a great, yet elusive, measure of effective work done to the powder. The model shows there is an optimum amount of strain required (around 1500) to attain a high-quality blend.

Lastly, Palmer et al. [75] shows a linear log–log relationship between Peclet Number (Pe) and strain, see Figure 17. Pe is a dimensionless number describing the ratio between advective and diffusive mixing in the system (see Equation (Equation 4)). To give an example, a high Peclet number would be expected to produce a narrow RTD. The model, at higher throughputs (and therefore, higher residence masses), showed reduced scattering and more linear behaviour, implying that the model may be favoured in predicting systems with high residence mass, whilst also implicitly suggesting that systems with low residence mass may present unstable micro-mixing capability and therefore may be unfavourable. Despite this, across all throughputs, it was seen that with reducing strain, the Peclet number increased.
(3)Strain=ϵ=MRT∗ω
where MRT is the mean residence time and ω is the agitator rotation rate.
(4)$$Peclet\;Number=Pe=\frac{advective\;transport}{diffusive\;transport}=\frac{uL}{D}$$
where *u* is the flow velocity, *L* is the characteristic length and *D* is the mass diffusion coefficient.
(5)$$Froude\;Number=Fr=\frac{centripetal\;force}{gravitational\;force}=\frac{\omega^{2}R}{g}$$
where ω is the blade rotation rate, and *R* is the geometrically relevant length-scale—i.e., the radial distance from the centre axis to the tip of a blade.

While, as outlined above, strain is considered to be beneficial to powder blending, it has also been shown that powders which experience high levels of strain (number of blade passes) are also more likely to be subject to triboelectric charging [13]. Naturally, the effect will present some interest in how electrostatics influence/interact with powder cohesion. Karner and Urbanetz [13] investigated the influence of pharmaceutical mixing in a batch mixer and found that particle size, the fraction of fine particles, and mixing volume were significantly impacting the charging behaviour of the powder, stating that small particles were more prone to generate higher charge density. How these findings would translate to a continuous blender represents a potentially interesting and valuable line of inquiry.

Beretta et al. [15,47] performed a two-part study on the triboelectric effect of charging powder via. powder transport in twin-screw feeding. This was followed by a study using relative humidity (RH) to subdue the electrostatic effect. Given that it is typical to maintain a RH between 30–60% in ISO-graded GMP environments [16] and that additional air moisture can increase the presence of liquid bridging, promoting cohesion [57], it becomes difficult to assess purely on the grounds of cohesion. The downside of controlling relative humidity is that it becomes difficult and expensive to maintain HVAC systems, especially as temperature and humidity fluctuate across the year. There are other options to consider, such as silication, outlined by Lumay et al’s study [57], which was shown to be capable of electrostatic dissipation and localised relative humidity control. Therefore, the use of silicates becomes a much cheaper and simpler option, despite what relative humidity may offer. However, this assumption is limited by current knowledge and awareness of the author(s) at the time of composing this review—novel literature may render this assumption moot.

Beretta et al. [15,47] stated that the charging forces are primarily recruited through particle–particle friction, while particle–wall friction did not directly contribute to charging. In addition, the authors made considerations for impacts on powder mixing; interestingly, the discussion touched on utilising these static forces to aid the mixing process [47]. Huang et al. [70] provide evidence of this, as dry powder coating is seen in a high-shear co-mill. Huang et al. [70] investigated the effect of dry powder coating of different APIs and excipients and found that the coating of fine colloidal silica improved both bulk density and FFC performance. SEM images displaying the silica coating are shown in Figure 19. Therefore, charge- or cohesion-based coating could provide an avenue to attain what Fan et al. [83] (see Figure 8) would describe as perfect ordered mixtures.

Lastly, alternative silicates—such as magnesium aluminosilicates (MAS)—have been trialled in some flowability studies; demonstrating an increase in flowability, without a constraint on mixing time and intensities, which is a limitation of several glidants [98]. Such materials could be used in place of the more traditionally used MPS grades for processes which may operate with different materials or mixing regimes.

### 3.3. Blending Summary

The scope of this topic is multidimensional, and therefore increasingly difficult to surmise without reducing the complexity of the situation; despite this, several trends—and avenues for future work—have emerged throughout this review.

Firstly, and perhaps most obviously, the literature strongly suggests that, regardless of differences between continuous blenders, cohesion provides a barrier to dispersion, and thus a barrier to achieving micro-mixing [78,90], which therefore decreases the quality of the mix. Cohesion has also been shown to exhibit complex behaviour at higher fill levels and lower rpms [75,78,90], where the particle–particle contacts are expected to be at their highest.

There is a common theme across the literature emphasizing the need to devise solutions to mitigate the effects of cohesion through the optimisation of processing parameters [43,45,75,88,99]. The literature suggests multiple manners in which this may be achieved, for example through alterations in system or agitator geometry, agitator rotation rate, feed rate, and various other factors. The differences in geometry for each blending system, however, means that each system has its own approach to solving the cohesion problem, thus making it difficult to generalise findings, and methodology, when attempting to understand the mechanisms and behaviour which explicitly govern cohesive blending. This is then exacerbated when considering that different formulations respond differently under the same conditions. An important task for future research is to define common parameters—e.g., dimensionless numbers—which can successfully generalise previous findings.

In the literature to date, several variables have been determined to affect the influence of cohesion on mixing. These include the Froude number (Fr, Equation (Equation 5)) or rpm, the magnitude of powder cohesivity, and the fill level of the blender. It has also been widely suggested that, due to cohesive particle–particle contact, a given system will require either a higher shear or more frequent agitation to break apart the additional cohesive forces. Strain (ϵ, Equation (Equation 3)) is defined as the mean residence time multiplied by the rotation rate of the agitator (RPM), and is a measure used to quantify the amount of work the agitator does per unit of space-time (τ) [43,75,82,88]. This quantity has been suggested as a valuable indicator of the degree of shear in a system and thus of blending success [43,75,82,88].

It has also been suggested that cohesion loses its significance, and complex behaviour, when rpm increases, implying that by running a continuous blender at high RPM, one may improve (micro-)mixing. However, the work of Van Snick et al. [43] suggested the need for a sufficiently large mass hold-up in order to ensure good macro-mixing. However, these seemingly contradictory requirements can potentially be reconciled by ensuring that RPM is high enough to minimise cohesion’s effect, thus improving micro-mixing, whilst altering the system’s geometry (or other system-specific variables) to ensure a suitable amount of hold-up mass is retained, thus retaining good macro-mixing.

Regarding future work, it would be interesting to see where the lines of significance lie for each of the three cohesion-dependent variables and their interactions. Such a study would most likely have to be conducted via DEM and should follow the suggestions left by Gao et al. [90]. In general, there is potentially significant value to be found in using numerical modelling methods such as DEM to address a number of the experimental observations discussed in this section.

#### A Perspective on Continuously Blending Cohesive Species

A final thought on continuous blending; is that the tablet’s powder formulation could be thought of as an emulsion, whereby work (in the form of strain) is given by the blender to shear through existing API clusters, reducing their size, and promoting the distribution and circulation of the new, finer API clusters around the bulk mass. The cycle then continues, by imparting even-more work and further distributing the API clusters. The remainder of the tablet’s formulation (which is mostly excipient) should, therefore, promote and stabilise the small API clusters, ensuring that they are maintained before they receive more work. Similarly, emulsions require thickeners and surface active agents to develop and maintain immiscible fluids [100]. Formulation development for CDC processes could look to use the same methodology, namely, through the exploitation of dry powder coating [70] and electrostatic charging [13]. However, considerations have to be made to ensure that, if the API is coated, it does not affect the therapeutic efficacy of the tablet.

## 4. Tabletting

### 4.1. Tabletting Introduction

The tabletting unit operation consists of three main steps:Die filling;Compaction;Tablet ejection.

The tabletting unit operation starts with the die-filling stage which fills the die with a consistent amount of powder mixture to achieve an adequate fill depth ready for the compaction step while also keeping good drug content uniformity [101,102]. The upper punch then lowers into the die increasing the compressional pressure applied to loose powder in the tabletting die; the powder consolidates as particles rearrange, filling void spaces and finally, as the availability of void spaces reduces, particles deform elastically, and ultimately plastically, before beginning to fragment [21] once the target pressure has been reached the upper die lifts releasing the pressure off the compact. Finally, the lower punch lifts the tablet to the top of the die ejecting the tablet. However, die filling and compaction are strongly influenced by cohesion whereas tablet ejection is not [103]. Qu et al. [103] study showed that additives such as silica which lowered cohesion did not lower ejection stresses effectively. Therefore, as this literature review focuses on cohesion, this section will only discuss die filling and compaction.

### 4.2. Tabletting Discussion

#### 4.2.1. Die Filling Introduction

As mentioned previously, die filling is a critical process step for tablet manufacture as mass and content uniformity are heavily dependent on the die-filling performance. If the die filling is not consistent, the mass and drug content uniformity is not accurate [104]. This will have an impact in the downstream processes as the quality attributes such as tensile strength and the dissolution profile will be affected [102]. The die-filling process is known to be affected by many bulk powder properties: flowability, cohesion, particle size, and morphology [101,102]. Many studies have shown a good correlation between these bulk powder properties and the die-filling performance [27,28,104]. Normally, dies are filled gravitationally using a shoe-die system where power is dispensed from a feed hopper which is connected to a box-shaped shoe which then moves over the opening of the die and allows powder to flow [38] (Figure 20). However, there are different mechanisms for die filling such as:Forced feeders [102] which use moving components (usually paddles) to feed the powder into the die. This is commonly used in turret presses [105] (Figure 20);Suction filling is where the lower punch also known as the ‘suction punch’ moves down creating a negative air pressure gradient promoting powder to move into the die, promoting the movement of the powder into the die [106] which removes the effect of air entrapment [38,107] (Figure 20).

Similar to Section 2 and Section 3, to have a good die-filling performance, the powder will need to have lower cohesive properties [3] These include larger particle size, spherical particle shape powder with lower surface energy. (Lower surface energy will mean that powder will find it more difficult to form strong interparticle bonds [103,108]), which all contribute to improved flowability [27,38,107]. To have content uniformity, the segregation must be kept to a minimum during the die-filling stage so as not to undo the work done during the mixing stage [107,109]. This section will focus on the die fill performance where the mass, segregation and fill depth are important to ensure the compression process will be successful.

#### 4.2.2. Die Filling Discussion

Cohesion has been seen to affect the die-filling performance in many studies as the flowability of the powder is reduced with increasing cohesivity [27,110]. As mentioned previously, CDC processes have fewer unit operations as the granulation step is bypassed. Granulation is where powder fines which have poor flowability are made into agglomerates. The powder fines adhere to one another commonly using a binder fluid (wet granulation) to form a larger multiparticle entity which are also known as granules [111]. The granulation step helps control key attributes such as flowability and the processability of the powder and therefore controls cohesion [112]. However, as the CDC process does not have a granulation step, the tablet formulation will need to have adequate flowability and tabletability [3] which limits the amount of tablet formulation that can be used with the CDC process. However, there are die-filling mechanisms which can improve the die-filling performance of poorer-flowing powders without the granulation step.

Wu [110] established ‘critical velocity’, which is defined as the maximum velocity which the shoe passes over the die with the die being filled after one pass [27]. The critical velocity is a common way to measure the flowability of the powder. A strong correlation is seen between critical filling velocity and mean particle diameter; specifically, an increase in particle diameter and a decrease in surface energy increases the critical velocity (Figure 21) [101]. This indicates that when the powder is more cohesive, the critical velocity is lower as the flowability is poorer, which means that cohesivity can have a negative impact on die-filling performance and cohesion control measures will be needed.

The effect of paddle speed in forced feeders was investigated by Goh et al. [107] which saw with a higher paddle speed there was an increase in die fill weight and reduced die variation. However, it was seen in other papers that the paddle speed only had an effect on the weight variability, not the fill weight itself [84] and only affected fair flowing materials such as MCC [113]. Therefore, a higher paddle speed can allow for cohesive powders to ‘flow better’ as they can essentially move as a solid block. On the other hand, in the study of Peeters et al. [114], results showed that the paddle speed affected the tensile strength. However, for the formulation with just microcrystalline cellulose, the tensile strength outcome was not affected by paddle speed. The only formulations that were affected were with magnesium stearate that saw a decrease in tensile strength with a higher paddle speed. Lubricants can have a negative effect on the tabletability if excessive shear force and higher levels of mixing occur. This is often referred to as ‘over-lubrication’ in the literature [115]. Therefore, the decrease in tensile strength is most likely due to these lubrication effects which is promoted with a higher paddle speed [116]. To understand the effects of lubrication in a forced feeder with higher paddle speeds, work should be conducted comparing the tensile strength outcomes of formulations with and without magnesium stearate. However, it should be noted that brittle materials are reported to not be as susceptible to the effects of lubricants as much as plastic materials [117]. Therefore, formulations with majority plastic powders will most likely be affected more heavily than formulations with a majority of brittle powders.

Another method to increase the flowability is by introducing air into the system, as flowing air gives a lubrication effect allowing particles to flow more easily with less resistance, increasing critical velocity [27]. The air prevents the percolation of fine particles, which reduces vertical segregation [118]. However, the presence of air can also create an adverse pressure gradient which resists the motion of particles, and pressure built up can further oppose the flow of the powder [27]. Suction filling has also seen positive effects with overcoming poorly flowable powders [38,106,109,119] and helps with the air entrapment which hinders powder flow as it creates a negative pressure gradient [38,107]. Furthermore, suction fill is generally known to lower the risk of segregation and improve the packing density [38]. This is seen in Figure 22 obtained from the study of Zakhvatayeva et al. [109]. The degree of segregation was measured using a sampling unit which separates the die into five sections; the process was repeated twice. The degree of segregation was calculated using Equation (Equation 6) where the segregation index indicates the level of uniformity in the powder blend in the die. Although the conclusion in Zakhvatayeva et al.’s [109] study is that suction filling improves die-filling efficiency and reduces segregation, in Figure 22 it is seen that gravitation filling with a fast die-filling velocity had a lower segregation index compared to suction filling. However, the slow die-filling velocity may not be representative of a real-life scenario in industry. Another interesting observation from Figure 22 is that with acetylsalicylic acid concentration of 10 percent, this had the worst overall segregation index across all cases which was explained as generally lower API concentrations are known to have a higher risk of segregation [120].
(6)$$Segregation\;Index\;=\;\sum_{i}^{n}\frac{c_{i}-c_{t}}{c_{t}}$$
where $c_{i}$ is the starting concentration of the blend and $c_{t}$ is the concentration of the analysed sample.

All these results show that there are promising ways to overcome the effect of particle cohesion on die-filling processes; however, it is often concluded that these results are highly dependent on the particles’ material properties [102,107]. Active pharmaceutical ingredients (APIs) are new and complex molecules that will often have unknown material properties [18]. Further studies should measure bulk properties and be able to understand what process parameters need to be utilised during the die-filling stage. APIs are known to be very cohesive and poorly flowing which gives the limitation that formulations for direct compression must have a maximum of 30% drug content [21].

A possible method to allow formulations with more than 30% drug content to undergo direct compression is spherical agglomeration/crystallisation, which will improve both processability and tabletability of the API. This is completed by changing the way the API is crystallised by either modifying the solvent addition rate and/or cooling rate [3]. Spherical agglomeration allows the API to crystallise spherically (Figure 23), which increases flowability and therefore lowers cohesion [3]. There are many promising studies using spherical agglomeration/crystallisation such as a study by Chen et al. [18] which saw an increase in flowability and therefore die-filling performance. Furthermore, the tabletability also improved (Figure 23) for the spherical agglomerated/crystallised produced using the quasi-emulsion solvent diffusion (QESD) method. The tensile strength was considerably higher compared to the ‘as received’ ferulic acid exhibits a needle-like elongated shape (Figure 23f). While spherical agglomeration/crystallisation is beneficial this should not be confused with the negative kind of agglomeration that occurs during secondary manufacture that can have a poor effect on blending, feeding and mixing. It is important to distinguish the two.

On the other hand, there are some disadvantages to spherical agglomeration/ crystallisation; for example, solvent selection is a difficult process and may be limiting due to the growing concerns of using more green solvents in industry [121]. This will need to be further researched and optimised in order to be used as a commercial technique [121,122]. Furthermore, the optimisation of process parameters required for the spherical agglomeration/crystallisation method is difficult and will need considerable work to scale up the processes [123]. Therefore, although this technique will help produce many more formulations suitable for direct compression for most drugs to take place, the preparation times at this current state will probably outweigh the benefits of just granulating. However, this technique is very promising and something future research should focus on.

#### 4.2.3. Compression Introduction

The aim of the compression process is to consolidate the powder mixture consisting of excipients and API into solid and cohesive compacts that can withstand the pressures of coating, packing, and shipping without fracturing. Therefore, the tablets will need to have a high enough tensile strength; however, if the tensile strength is too high this will typically lead to low porosity [30]. If the porosity is too low this could negatively affect the dissolution rate required to deliver the drug to the patient [124]. There are many aspects that contribute to the tablet’s strength, such as moisture content [125], compression speed [126], and granulation [127]. However, as this literature review is focusing on the effect of cohesion and assumptions can be made that the tabletting process is completed under strict humidity controls [100], this section will just be focussing on surface energy and particle size and shape. These three particle properties contribute heavily to the magnitude of cohesion [128]. Although cohesion effects for other process operations in the tabletting manufacturing process are known to have a mostly detrimental effect (see Section 2 and Section 3), in contrast, cohesion properties are important to the compression as cohesivity helps produce a strong solid compact. Therefore, it is important to understand how the cohesion properties contribute to the tabletting stage so potential compromises can be taken to help the formulation perform well throughout the manufacturing process.

#### 4.2.4. Compression Discussion

The effect of particle size on powder compaction has been the focus of many papers where it is generally understood that smaller particle sizes form a stronger tablet [29,30,31,32,129]. Smaller particles have been found to have a larger specific surface area where interparticle bonding can take place resulting in stronger tablets [130,131]. However, smaller particles are typically more cohesive and have poorer flowability which will affect die filling (Section 4.1).

Particle shape is commonly linked with particle size effects in tabletting and has similar effects on the flowability and cohesivity of the powder [132]. A more irregular shape will exhibit poorer flow properties and be more cohesive compared to regular or spherical-shaped powders [133]. Additionally, irregular-shaped powders have been shown to form a stronger tablet, which was demonstrated by Johansson and Alderborn’s [30] study. The authors found that tablets containing irregular shaped MCC granules had a higher tensile strength than spherical-shaped pellets (Figure 24). Both granules exhibited plastic deformation as the dominant deformation mechanism, but at higher pressure, there was some fragmentation/attrition which resulted in a closer pore structure. Šimek et al. [133] investigated the effects of different particle shapes (Figure 24) of a brittle material, paracetamol. Their findings were similar to Johansson and Alderborn’s [30], where more irregular shaped paracetamol had increased compressibility shown with the Heckel plot. The Heckel plot is a commonly used equation and plot to help understand the densification of the powder during compaction and allows interpretation of how plastic the powder is which is taken from the yield pressure which is derived from the reciprocal of the linear region gradient of the curve. The lower the yield pressure, the more plastic the material is [134]). Figure 25 compares this to spherical-shaped paracetamol.

Although the general trends of particle size and shape’ effects on tabletting is mentioned in the previous section, there are some conflicting results regarding how they affect the compressibility and therefore tablet strength. This can be differentiated by material properties or the dominating deformation mechanism of the powder. Brittle materials such as lactose and paracetamol deform by fragmentation [135]. In studies, the strength of tablet consisting of brittle materials have been seen to be affected by particle size [29,128,136,137] and shape [138]. Skelbæk-Pedersen et al. [137] found that larger brittle powder particles such as lactose and calcium hydrogen phosphate dehydrate (DCP) fragmented more. In Figure 26, with increasing particle size in DCP and lactose, the peaks are more flatlines for the lines corresponding to the higher compressional pressure which means the particles fragmented into smaller particle sizes more readily. This agrees with De Boer et al. [29] Sun and Grant, [130] and Skelbæk-Pedersen et al. [137]. The more extensive fragmentation allows the void spaces to be filled with the smaller particles which further densifies the powder bed to reduce its negative influence of tensile strength. Nonetheless, smaller particles have higher tensile strength outcomes after compression [130]. Surprisingly, in contrast, Almaya and Aburub [117] found that dibasic calcium phosphate dihydrate did not have a difference with different particle sizes. Suggestions into why this may be occurring will be discussed below.

Plastically deforming powders are known not to be affected by particle size [117,128,136] due to the way they deform. McKenna and McCafferty [128] found that different sizes of microcrystalline cellulose (MCC) did not affect the tensile strength. However, Herting and Kleinebudde [116] found contradicting results that decreasing particle size of MCC and theophylline resulted in higher tensile strength. Wunsch et al. [132] found that there was an increase in tensile strength with decreasing particle size for all materials investigated (MCC and vildagliptin) (Figure 27) However, with vildagliptin, internal defects such as cracks during elastic recovery may have occurred resulting in large error bars. No fundamental understanding of these differing results is given in the papers, particularly regarding the contradictions with past literature.

Despite these contradictions, which need to be further understood, a common conclusion can be made. Many papers agreed that, when the change in particle shape and size saw an increase in tensile strength, this was due to the specific shape or size which resulted in an increase in area of contact between particles under a compressional pressure [29,30,67,130,132]. Wünsch et al. [132] found that the reduction of initial particle size changes the deformation behaviour and therefore the effect of particle size is dependent on the specific material. With increasing initial particle sizes, the specific plastic energy rises but the specific elastic energy and elastic recovery is not affected as much. Therefore, when reducing sizes of materials, this could affect their mechanical response to the exerted compressional pressure. Furthermore, crystal structures may have important information about mechanical properties of the powder which is also seen in Sun and Grant’s [130] study which saw two different polymorphs of sulfamerazine exhibit different compaction properties (Figure 28) Similar results were seen in Upadhyay et al.’s [139] study, where it was found that the different polymorphs of ranitidine hydrochloride exhibited different compressibility results. In a later study using the same polymorphs, polymorph I had better tabletability compared to polymorph II at all size fractions [140].

Future studies should pay more attention to crystal structure and understanding the corresponding mechanical properties of the materials being investigated before and after they are processed to a certain particle size which could affect the mechanical properties of the powder. For example, Simek et al. [133] modified their paracetamol samples in ethanol and changed the stirring speed to achieve the desired particle shape (Figure 25). Understanding these discrepancies could allow potential compromises to be made. For example, if a certain material is very cohesive but the particle size is independent of the tabletability properties, a larger particle size could be used to reduce cohesivity while not affecting the tabletability of the tablet formulation.

Surface energy is the excess energy on the surface compared to the bulk [141] which is affected by a number of intermolecular forces such as van der Waals [142]. A more cohesive powder will have a higher surface energy with interparticle bonding that will hinder the powder flow [143]. However, it is well documented that surface energy has a profound effect on producing successful tablets after powder compaction. Many results have suggested higher surface energy leads to stronger bonds forming between powder particles, which form higher tensile strength tablets [33,34,35,36]. For example, Wünsch et al. [132] suggested that formation of hydrogen bonds between MCC particles is why there was a considerable increase of tensile strength compared to ibuprofen and vildagliptin, as seen in Figure 29. However, the way that MCC, ibuprofen and vildagliptin deforms was not considered in this discussion point.

To intrinsically measure the effect of surface energy of the powder on the tensile strength is immensely difficult. Fichtner et al. [144] and Chen et al. [145] have attempted to change the surface energy of the powder by dry coating with polysorbate and Aerosil 200 respectively. This showed a decrease in tensile strength with the decrease of surface energy in both studies. However, the effect of changing particle size was not well documented and as previously stated in Section 4.2, depending on the specific material, particle size is another factor that changes the tablet outcome after compression. Ho et al. [146] counteracted this issue by changing the surface energy without changing any other powder particle property by modifying the functional groups on the surface of the powder without disrupting the surface morphology and particle size by silanisation. This treatment profoundly decreased the tensile strength of the tablet as the surface energy was reduced [147].

Another method, which was also mentioned in Section 2.1 to decrease the surface energy of a powder and improve flowability is to add a glidant, which is seen to increase the flowability by reducing cohesion [6]. How these additives affect the tabletability of the powder formulation was looked at in many different studies. Magnesium stearate (MgSt) is known to improve flowability [148]. However, it has a detrimental effect on the tablet strength especially when magnesium stearate is mixed for too long and vigorously. This is an effect which is known as ‘over-lubrication’ [116,149]. The MgSt is theorised to create a thin hydrophobic layer around powder particles preventing bonding between the powder particles to take place and therefore decreasing the tensile strength [149]. Furthermore, as MgSt is hydrophobic, the powder being ‘over-lubcricated’ will effect the dissolution [150]. On the other hand, MgSt has excellent lubrication properties and is commonly used in tablet formulation as a lubricant [144] rather than a glidant.

A glidant that is discussed frequently in literature is silicon dioxide. Apeji and Olowosulu [67] used talc and colloidal silica which appears to have had a negative influence on the tensile strength of the tablet. This paper did not have any comparison to a control data without any glidant to compare with. What the previous papers mentioned do not include is whether they are using hydrophobic or hydrophilic silicon dioxide. Kunnath et al. [151] looked at both hydrophilic and hydrophobic nano-silica and their impact on tablet tensile strength and dissolution (Figure 30). The nano-silica was dry coated onto the API with similar methods to those implemented by Chen et al. [145], mentioned previously. This saw an increase in tensile strength for both hydrophilic and hydrophobic nano-silica and did not see an effect on the dissolution profile. The possible explanation of this is that the dry coating of API led to deagglomeration of the powder and increased the total contact area where interparticle bonding can take place. On the other hand, in Mužíková et al.’s [66] study, when colloidal silica (Aerosil 200 and 255) was added, this saw a decrease in tensile strength, therefore the disintegration time also decreased. The max compressional pressure used to produce the tablets was very low (26.37 MPa) compared to what is normally used (200–250 MPa). However, Wunsch [132] found that it is the number of bonds which determine the strength of the tablet, not the bonding strength. Similar to the particle size conclusion, the fundamentals of the mechanical properties and therefore deformation properties of the powder are vital to understand as enhancing the bonding area may be more important than increasing the surface energy. Furthermore, this could mean a compromise can be taken on surface energy which will lower the cohesivity of the powder but would not affect the tablet outcomes if the contact area between particles is sufficient enough to gain the number of bonds needed. This further points to the importance of understanding how the powder deforms and how that effects and governs the contact area between particles [19].

There have been promising results for silicon dioxide alternatives such as magnesium aluminosilicate (MAS) and tricalcium phosphate (TCP). Both of these novel glidants have a positive effect on flowability [98]. However, there is just a single study investigating the tabletability of MAS, and none with TCP; in which, Hentzchel et al. [152] compared the tabletability of MAS with other silicates, when mixed with MCC of different concentrations. MAS was the only silicate that did not decrease in tablet strength, with increasing concentration, which is a positive result. Although the tablet strength is not affected by MAS, a study on how these novel glidants affect the dissolution rate at different concentrations will need to be conducted, as this is a critical quality attribute of the tablet. A study by Khunawattanakul et al. [153] saw the MAS in a tablet film coat increased the dissolution time, suggesting a balance will be required to ensure the right concentration to aid flowability is achieved, without negatively affecting dissolution. Although Tran et al. [98] shows that MAS aids flowability, this will need to be further investigated into whether MAS does have an impact on the feeding and blending parts of the CDC process. As far as the authors are aware there are no extensive studies that analyse the effect of these alternative glidants on these unit operations, essentially showing that additional studies are required to understand the complex interactions between not only CDC processing but the resultant in vivo performance of the tablet. The aforementioned publication Tran et al. [54] takes a perspective of using a range of measurement techniques to inform quantities or types of flow modifiers, practically taking a quality by design approach. A similar methodology could be applied in terms of glidants (namely, MAS) and their subsequent impact on tablet quality CQAs such as tensile strength of binary mixtures with excipients other than MCC.

### 4.3. Tabletting Summary

In this section, cohesion was split between two main contributing factors, particle size and shape, and surface energy, where smaller particle size and higher surface energy generally resulted in a higher magnitude of cohesion [24]. Although there are some contradictions seen in the particle size section, generally the tabletting process needs a good level of cohesion to be successful. However, spherical agglomeration is a promising technique that can lower the effects of cohesion seemingly without negatively affecting the tabletability [3,122]. More research will need to be conducted to optimise the solvent selection process [123].

Furthermore, both particle size and shape and surface energy research pointed towards the fundamentals of the mechanical properties of the powder. Particle size and shape discrepancies may be due to how the particle size and shapes were manipulated and how the crystal structure and therefore the material properties have changed. Additionally, it was found in Wunsch’s et al. [132] paper that it is more the number of bonds rather than the strength of individual bonds that influences the overall tablet strength. Therefore, a higher contact area between particles to allow for a higher number of bonds could be more important than having higher surface energy powders to obtain stronger tablets.

The addition of glidants which increased flowability are seen to have contradicting results from commonly used silicon dioxide discussed in the previous section. On the other hand, promising initial results from modern alternative glidants, such as magnesium aluminosilicate (MAS), showed an increase in flowability and no change in tensile strength of the tablet regardless of concentration [98]. However, more studies need to be completed to scope what kind of an effect MAS has on dissolution and the tensile strength of binary mixtures with different commonly used excipients with different deformation properties than MCC.

## 5. Summary Table

Table 2 summarises the effect cohesion has on the unit operation and what control measures are outlined in the discussion. How the control measures from one-unit operation can affect another downstream has yet to be discussed. This section will outline how some cohesion controls that have positive effects on unit operations can promote issues in downstream unit operations in the continuous direct compression line.

The consistency and accuracy of which powders are dispensed from a feeder will directly influence the downstream unit operations [43,45]. Therefore, it is of utmost importance that understanding is gained on how to maintain a consistent feed rate for very cohesive materials [2,41,55]. The use of silica shows great promise, as it seemingly both improves flowability and reduces electrostatic charging, leading to better screw filling and thus resulting in improved feeding consistency [5,47,57]. Lumay et al. [57] also showed silica’s ability to influence the thixotropic behaviour; in future work, it would be interesting to see a development on this in the context of LIW feeding. However, the addition of silica has potentially negative effects on tabletting [66,67], while its effect on blending has not been explicitly described. Bridging/arching within the hopper volume has been shown to suffocate the flow of powder into the conveying volume. As a result, both additives and refilling strategy show ideas for reducing the processing variability [2,41,55,65]. Furthermore, using DEM for complex agitator design would further help the development of arching minimisation.

Despite the benefits of silication, attention should be paid to the end-to-end impact of formulation additives. For instance, nano-silicas were seen to have a lesser impact, compared to MPS, on the tablet strength, but greatly improved the flowability [57]. Alternatively, novel glidants, such as magnesium aluminosilicates, show promise in minimising the respective trade-off of improving flowability and affecting the tablet’s tensile strength [152].

Blending seeks to intimately mix all of a given formulation’s constituents and deliver them to the final, tabletting stage of the process with minimal variation [45,154]. Cohesion, in most cases, serves as a direct barrier to blending, as it provides resistance to intimate mixing [75,78,90]. Thus, higher levels of agitation are required to break apart these cohesive clusters, but systems must devise their own method to balance both the micro and the macro aspects of mixing [43,88]. In addition, complex bulk behaviour occurs when there is a moderate amount of cohesive particle–particle contacts [90]. Thus, it is of interest to develop a greater understanding of the mechanisms which govern this complex bulk behaviour. Moreover, despite the viewpoint to increase RPM to deal with cohesion, it is not understood what these new shear environments look like, and whether they have an impact on tabletability.

Spherical Agglomeration, which was discussed in Section 4.2, is a very promising technique to control the amount of cohesivity APIs add to tablet formulations without compromising tabletability to the same extent as the addition of glidants and lubricants which is beneficial to the whole CDC process. Further work will, however, need to be carried out to improve the efficiency of selecting the right solvent and process parameters [123]. Process parameters especially will need to be focussed on as some parameters are highlighted in studies that may be difficult to scale up. However, there have been good strides looking into the scale-up process and making spherical agglomeration/crystallisation continuous [155,156]. Ensuring the API crystals are not too large will be important to ensure content uniformity within tablets [123].

**Table 2 pharmaceutics-15-01587-t002:** Summary table with all the unit operations discussed in this literature review and their respective cohesion effects and controls. Note: Cohesion compromise is denoted with a ⋄, future work with ∘, considerations Δ.

Unit Operation		Cohesion Effects Results in …	Cohesion Control or Compromise ${}^{⋄}$	Future Work ${}^{\circ }$/Considerations ^Δ^
Feeding		Inconsistent screw filling	Modifying bulk density and cohesive particle-particle contact through additives, such as Silica [5,47,57]	Understand and be able to modify thixotropic powder ${}^{\circ }$ behaviour
			Increase/Decrease conveying screw speeds [57]	
		Bridging/Arching	Refill Strategy [41,65]	DEM simulations exploring optimum hopper agitator design ${}^{\circ }$
			Agitator Design	
Blending		Reduced micro-mixing	Increase in rpm (Fr) balanced by changes to system geometry to maintain macro [75,78,90]	
		Complex bulk behaviour, which results in poor mixing	Modify the three contributing factors: Fill level, Magnitude of Cohesion, and Froude Number [43,75,88,90]	
			Mechanistic understanding of how these factors interact [90]	
Tabletting				Find a more efficient way to find the right solvents ${}^{\circ }$
	Die filling	Incomplete die fill and weight [102,110]	Spherical agglomeration [3]	Understand how spherical agglomeration will affect other CDC unit operations (feeding and blending) ${}^{\circ }$
				Investigate the possibility of a scale-up method as the will be hard as the process is difficult and time consuming on small scale ${}^{\circ }$
			Paddle speed [107]	Overlubrication can occur if the paddle speed is too high [115]. A compromise will need to be taken to ensure an increase of paddle speed will not effect tablet strength ^Δ^
			Introduction of air in the system acting like a lubricant [27]	As this can create an adverse pressure gradient which will then resist the motion of particles. Find the ranges when air introduction is beneficial by Quality by Design experiments ^Δ^
	Compaction	Better tabletability; generally lower particle size and higher surface energy results in stronger tablets but can have bad effects on dissolution if the tablet strength is too high [157]	Nano-silica and other novel glidants (magnesium aluminosilicate) can help promote better flowability and has little to no effect on tablet strength ${}^{⋄}$ [151]	Understand the impacts of different types of nano-silica. Magnesium aluminosilicate will need to be tabletted with other common excipients and dissolution testing will to be conducted to assess the limitations ${}^{\circ }$
			Number of bonds is more important than the strength of the bond could be able to compromise lower surface energy/cohesivity if there is sufficient contact area gained during compaction ${}^{⋄}$	Understanding crystal structure fundamentally and how that changes material properties. This may be what is causing discrepancies regarding particle size and the tablet strength outcome ${}^{\circ }$ [130]

## 6. Conclusions

This literature review consolidates the impact of powder cohesion and mitigation for each of the unit processes in CDC, whilst considering the consequences of upstream/ downstream processing. For a tablet formulation to be viable for the CDC process, the formulation must possess both suitable flowability *and* tabletability [19,25]. Cohesion is known to cause powders to resist flow [23,24] which has a negative impact on the performance of the feeding [25], mixing [85,87], and die filling [3] unit operations involved in the CDC process whereas, for the compression stage, cohesivity is desired [29,30,33,35].

Understanding the cohesive nature of APIs and excipients and then manipulating them, where possible, is useful for improving the overall performance of the CDC process. The use of additives such as glidants and lubricants may aid the flow of powders, thus improving the feeding [6], mixing [37] and die filling [110] performance; however, these additives can have a negative impact on compaction [29]. Flow variability should be managed (specifically during the feeding stage) as any perturbations will be propagated into successive unit operations [2,158].

Despite the volume of literature in this area, there remain many important open questions and thus valuable future research avenues:Silication has benefits similar to glidants and lubricants as it can lower the cohesive behaviour of the powder and therefore increase flowability [6]. However, there are contradictions on whether silication negatively affects tabletability; this will need to be further investigate [67,145]. Alternative glidants, e.g., magnesium aluminosilicates, are a promising option however, more work needs to be carried out to understand the impact on tablet performance.Discrete Element Method (DEM) modelling, allowing researchers to create a digital twin(s) of an existing experiment and conduct statistical analysis on the simulated powder behaviour would allow researchers to gather metrics to quantify mixing/feeding performance (as discussed in Escotet-Espinoza et al. [80]) that were not accessible experimentally.Utilising triboelectric charging for the blending stage: static surface charge is developed on the surface of particles due to the strain given to the powder during mixing/transport [47], potentially allowing a formulation to be altered to promote attraction/repulsion between constituent species in order to gain a more-ordered well-mixed system.Spherical agglomeration/crystallisation can improve the flowability and tabletability of APIs, which is normally the most cohesive component of the tablet formulation [3,18]. However, further work needs to be performed to improve the solvent selection and process parameters to allow for scale-up [123].The many discrepancies in the literature regarding the manner in which particle size affects the tabletability of the powder should be comprehensively addressed, as this represents an important gap in the fundamental understanding of powder compression [39,128]. Understanding which types of powders are affected by particle size and which are not could lead to compromises to have larger particle sizes to lower cohesivity and therefore improve flow while not affecting the tabletability.

## Figures and Tables

**Figure 1 pharmaceutics-15-01587-f001:**
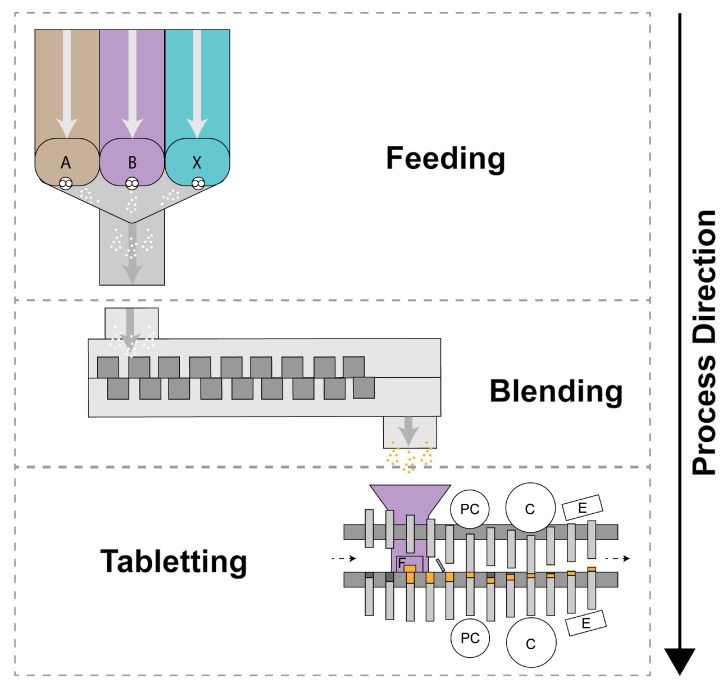
Schematic of the CDC process. Feeding: (where **A**, **B**, & **X**; are example labels of the formulation’s constituents) powders are fed into the blending stage. Blending: Shows the constituents entering and being blended by the agitator, before entering the hopper of the rotary tablet press. Tabletting: (where **F**, **PC**, **C**, & **E**; corresponds to the Die Filling, Pre-Compression, Compression and Ejection, respectively) the final formulation fills the die cavity, undergoes compression forces—forming the tablet—before being ejected from the die.

**Figure 2 pharmaceutics-15-01587-f002:**
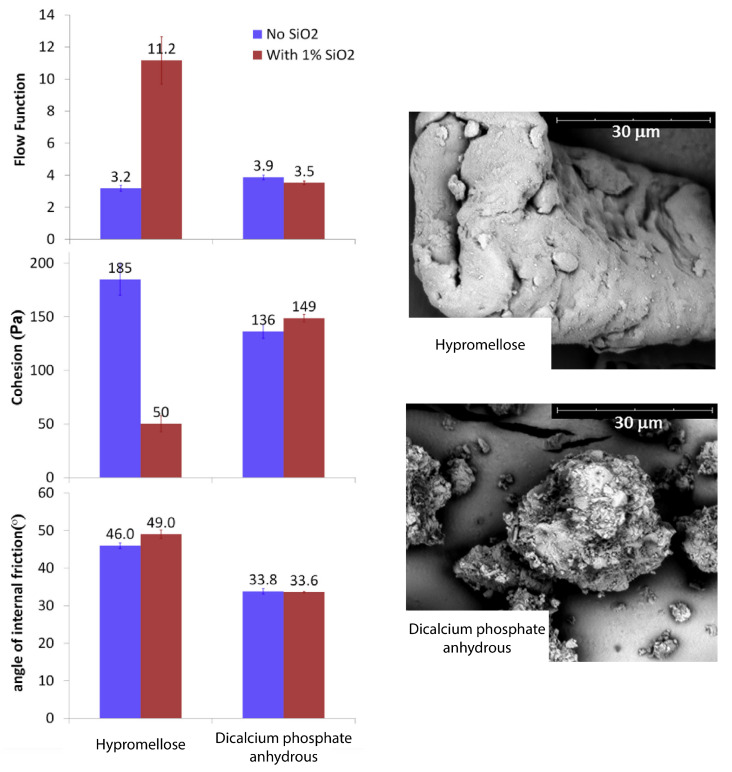
(**Left**): Graphs highlighting the difference in characteristics (Flow Function (FFC), Cohesion, and angle of internal friction) of the two excipients: Hypromellose and Dicalcium Phosphate Anhydrous with and without colloidal silica. (**Right**): Scanning Electron Microscope (SEM) of the two excipients mixed with 1% *w*/*w* colloidal silica. Adapted from Leung et al. [5].

**Figure 3 pharmaceutics-15-01587-f003:**
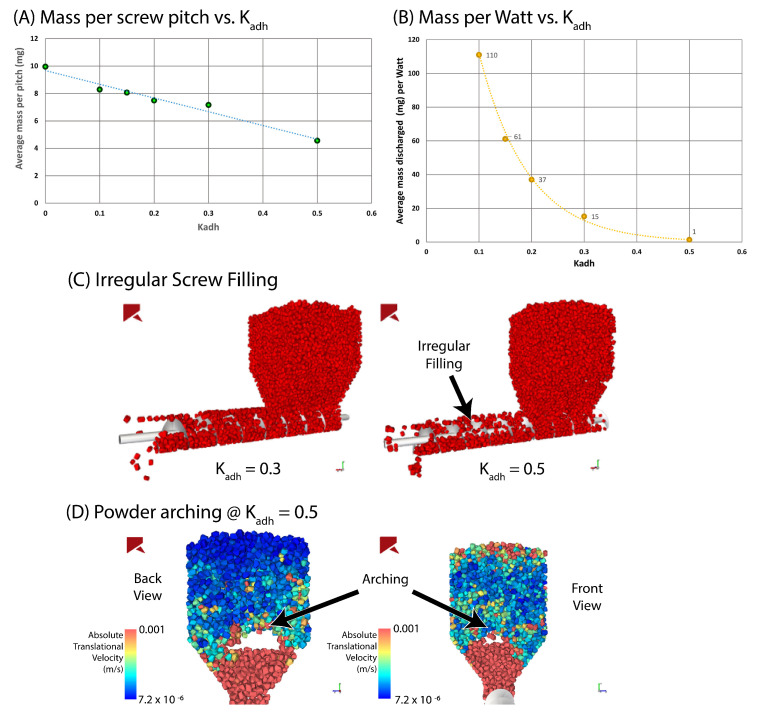
(**A**): Mass per screw pitch as a function of cohesive stiffness ($K_{adh}$). (**B**): Mass per Watt as a function of cohesive stiffness ($K_{adh}$). (**D**): Visualisation of the powder arching/bridging within the hopper volume. The images show both the back and front of the hopper from two different simulations using the same $K_{adh}$. (**C**): Irregular filling of the conveying volume due to cohesive stiffness ($K_{adh}$). The images compare $K_{adh}$ = 0.3 and $K_{adh}$ = 0.5. Adapted from Lopez et al. [55].

**Figure 4 pharmaceutics-15-01587-f004:**
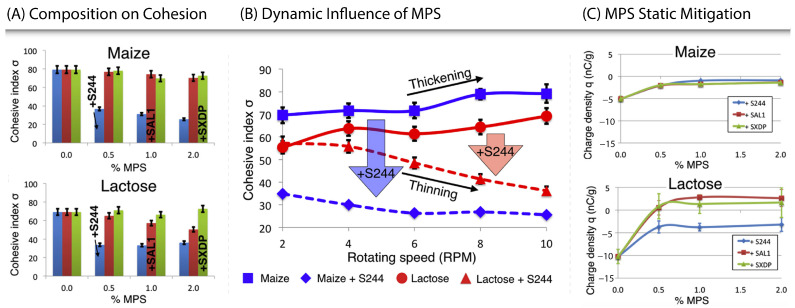
(**A**) Comparing the differences in the dynamic cohesive index (σ) with the addition of different types and weight percentages of Mesoporous Silicates (MPS). (**B**) Dynamic influence of MPS on the cohesive index, demonstrated through plotting the cohesive index over different rotating speeds. (**C**) Comparison of Charge Density (nC/g) in two excipients and resultant performance with the addition of 2% weight silicates. Where: +S244 is Syloid^®^ 244FP, +SAL1 is Syloid^®^ AL–1FP, and +SXDP is Syloid^®^ XDP3050. Adapted from Lumay et al. [57].

**Figure 5 pharmaceutics-15-01587-f005:**
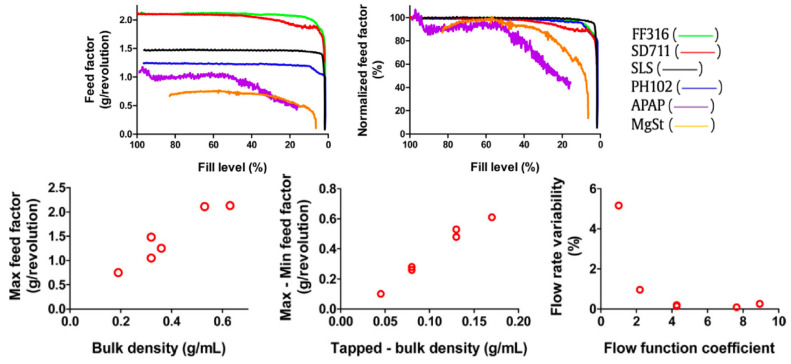
(**Top**): Two graphs detailing the feed factor throughout the volumetric empty, with absolute values (**left**) and normalised values (**right**), the key indicates the material used. Where: FF316 is Lactose Fast Flo^®^ 316, SD711 is Sodium Croscarmellose Ac-Di-Sol^®^ SLS is Sodium Lauryl Sulphate, PH102 is MCC Avicel^®^ PH-102, APAP is Acetaminophen, and MgSt is Magnesium Sterate. (**Bottom**): Three graphs compare feeder performance to bulk characteristics. Adapted from Van Snick et al. [43].

**Figure 6 pharmaceutics-15-01587-f006:**
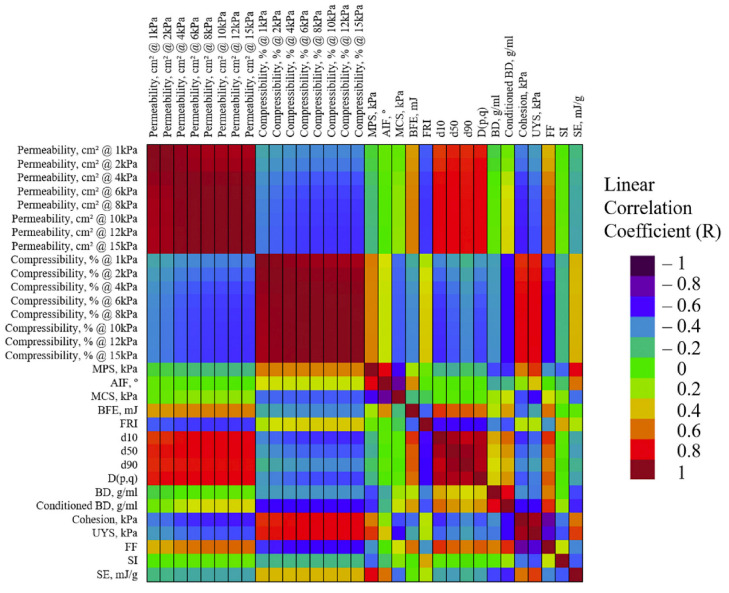
A correlation heat map acquired, with permission, from Escotet-Espinosa [62], detailing the magnitude of linear correlation (R) between bulk powder characteristics and regressed feed factor parameters.

**Figure 7 pharmaceutics-15-01587-f007:**
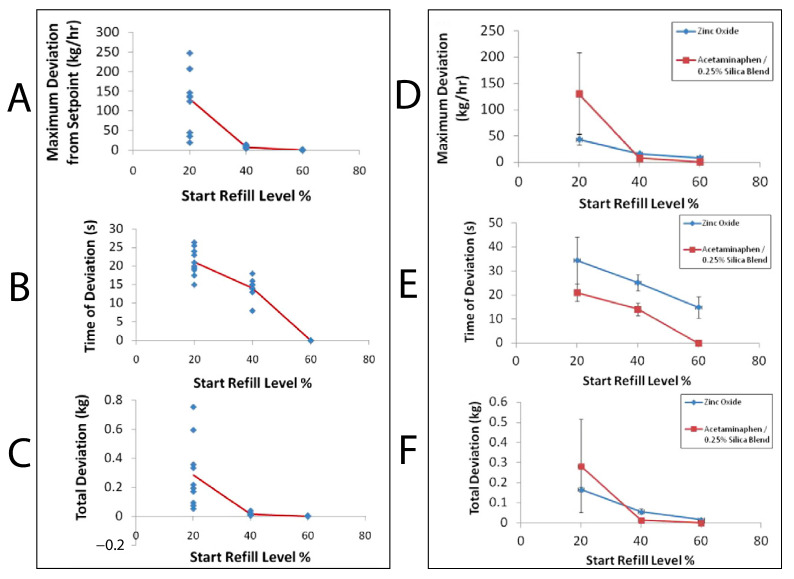
(**A**–**C**) Shows the feeder performance for just semi-fine APAP & 0.25% silica over 3 refill levels, each with 10 manual refills. (**A**) Maximum deviation from setpoint (kg/h), (**B**) Time of deviation (s), and (**C**) Total deviation (kg). (**D**–**F**) Similarly, shows the comparison between zinc oxide powder and semi-fine APAP & 0.25% silica blend. (**D**) Maximum deviation from setpoint (kg/h), (**E**) Time of deviation (s), and (**F**) Total deviation (kg). The deviation is defined as the sum of the surplus powder delivered during refill. Adapted from Engisch & Muzzio [41].

**Figure 8 pharmaceutics-15-01587-f008:**
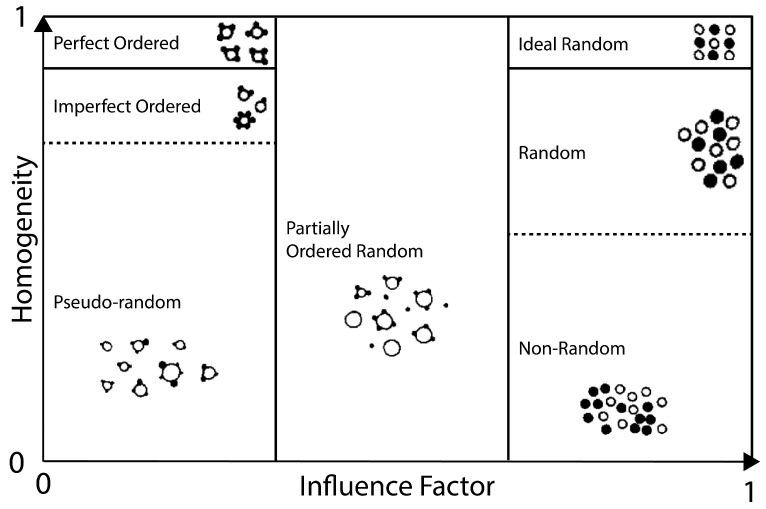
Cohesive mixtures described by homogeneity vs influence factor (Equation (Equation 2)). Adapted from Fan et al. [83].

**Figure 9 pharmaceutics-15-01587-f009:**
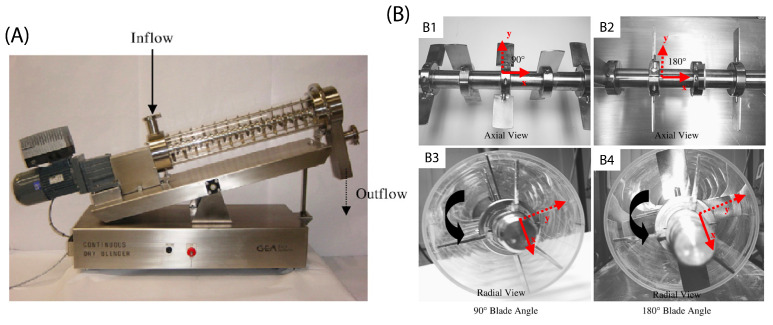
Example linear blender used in Portillo et al. [91]. Profile view of the blender (**A**) view of the agitator shaft and blade angle orientation (**B**).

**Figure 10 pharmaceutics-15-01587-f010:**
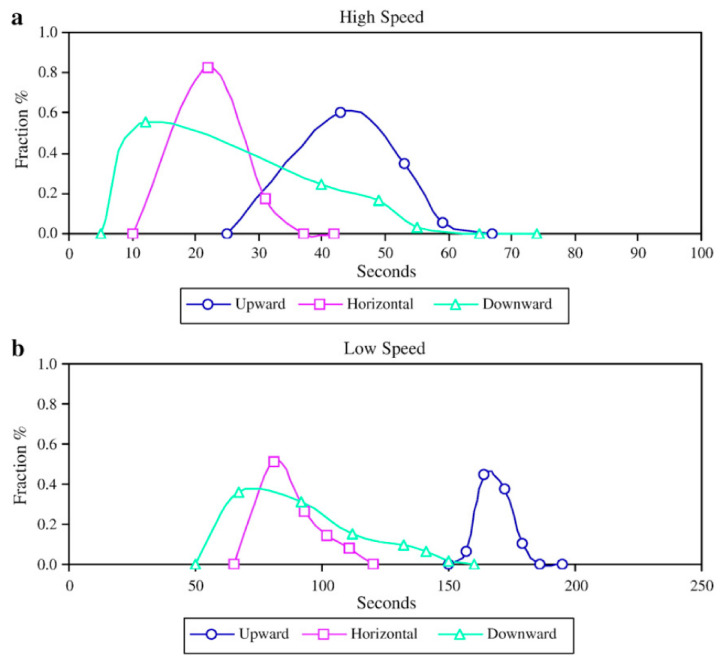
Acetaminophen residence time distribution plots (gathered with permission) from Portillo et al. [91], demonstrating the effect of blender volume inclination and RPM on RTD; (**a**) 78 RPM, (**b**) 16 RPM.

**Figure 11 pharmaceutics-15-01587-f011:**
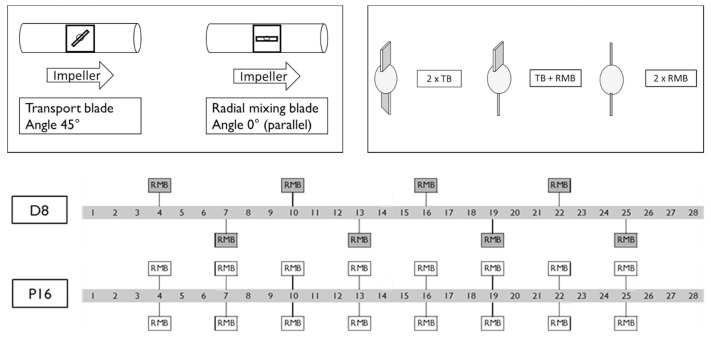
Blade configurations refer to a series of different blade orientations along the agitator’s axis. (**Top left**): side-by-side comparison of transport blade and radial mixing blade (RMB). (**Top right**): example of a blade combination that would take a position along the agitator’s axis, denoted by a number in the configurations D8 and P16 below. D8 and P16 depict blade configurations; all positions that are not labelled as RMBs are transport blades, and each collar position sees the blade combination rotate 60${}^{\circ }$ clockwise to the direction of the impeller. Adapted from Van Snick et al. [43].

**Figure 12 pharmaceutics-15-01587-f012:**
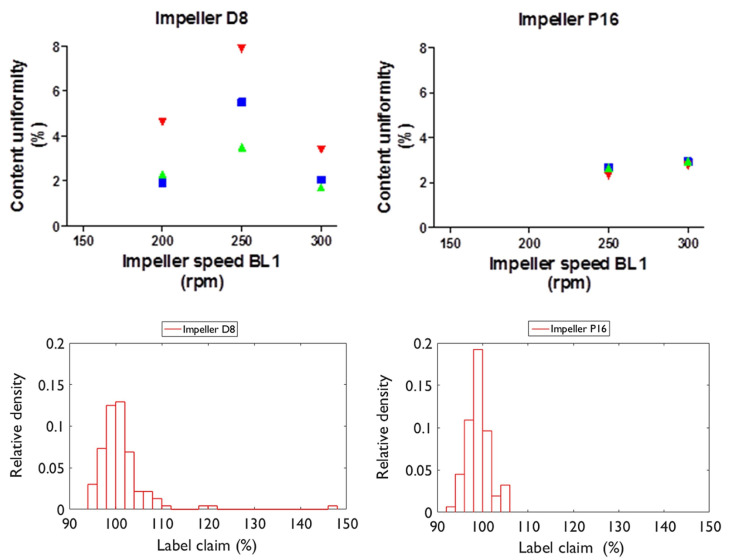
Tablet uniformity corresponding to the blade configurations seen in Figure 11 (D8 and P16); Upper graphs: content uniformity as a function of impeller speed and mass rate (green triangle = 24 kg/h, blue square = 30 kg/h, red triangle = 36 kg/h). Lower graphs: relative density distribution based on the target API dosage at the centre point of the experimental design (250 rpm and 30 kg/h). Gathered with permission from Van Snick et al. [43].

**Figure 13 pharmaceutics-15-01587-f013:**
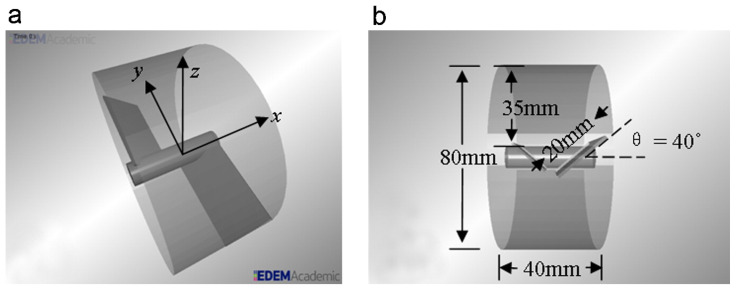
Periodic section of a continuous blender including two blades (**a**) Isometric View, (**b**) Side profile view with dimensions. Gathered with permission from Gao et al. [95].

**Figure 14 pharmaceutics-15-01587-f014:**
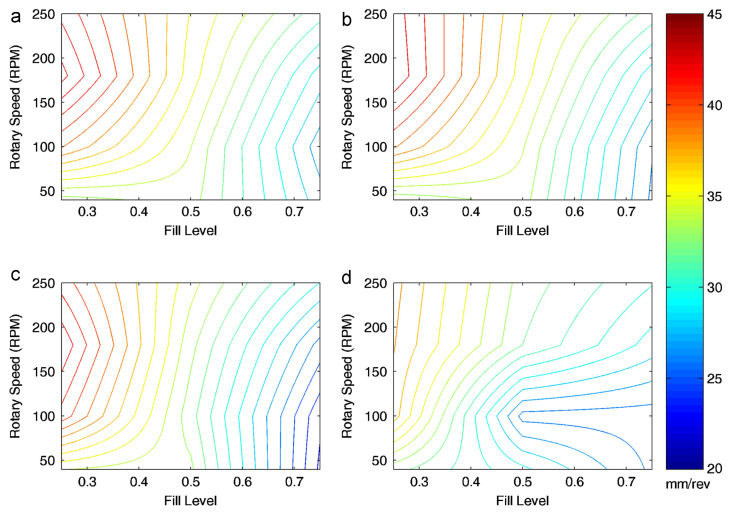
Contour plots of normalized mean axial velocity: (**a**) control, (**b**) lower density, (**c**) larger particle size, and (**d**) cohesive particles. Gathered with permission from Gao et al. [95].

**Figure 15 pharmaceutics-15-01587-f015:**
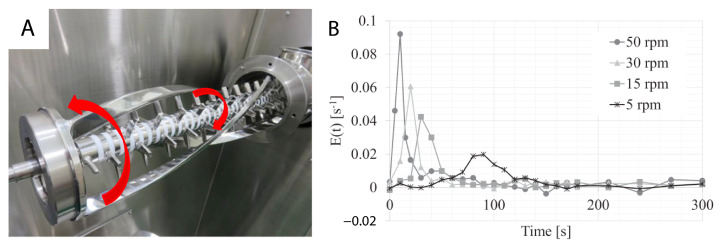
(**A**) Impeller and scraper continuous mixer with arrows detailing rotation direction, (**B**) APAP RTD spike response. Adapted from Tomita et al. [97].

**Figure 16 pharmaceutics-15-01587-f016:**
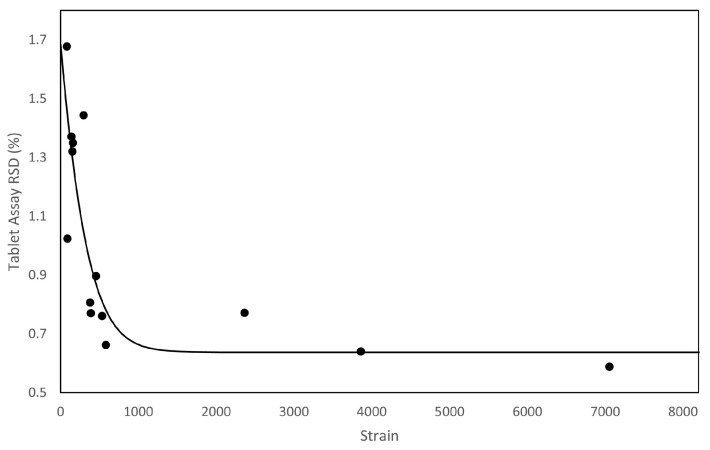
Tablet assay RSD plotted against Strain (see Equation (Equation 3)), an exponential decay model is fitted to the data. Gathered with permission from Palmer et al. [75].

**Figure 17 pharmaceutics-15-01587-f017:**
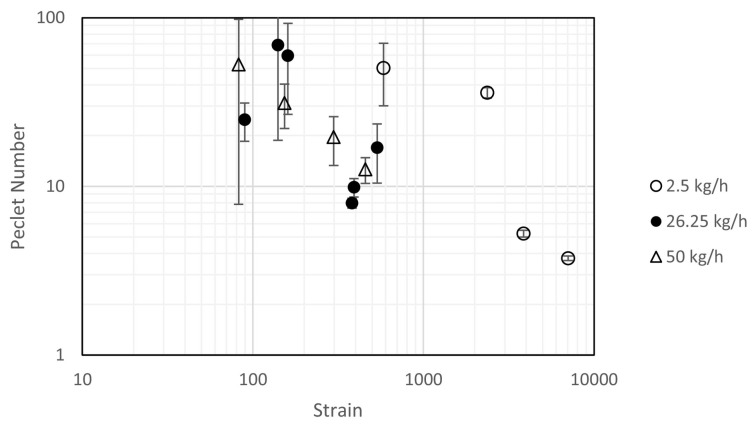
Peclet Number (Pe) and Strain relationship plotted on a log–log graph, grouped by throughput. Gathered with permission from Palmer et al. [75].

**Figure 18 pharmaceutics-15-01587-f018:**
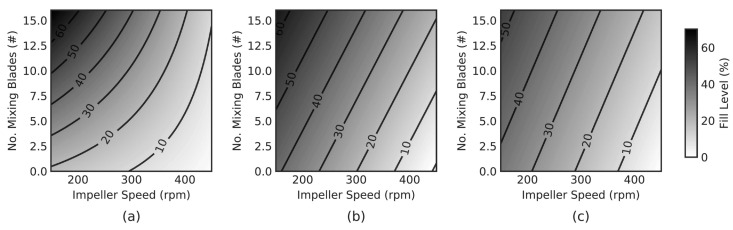
Contour plots mapping the impeller speed, number of mixing blades, and fill level at a throughput of 26.25 kg/h (**a**) Micronized APAP, (**b**) powder APAP, (**c**) special granular APAP. Gathered with permission from Palmer et al. [75].

**Figure 19 pharmaceutics-15-01587-f019:**
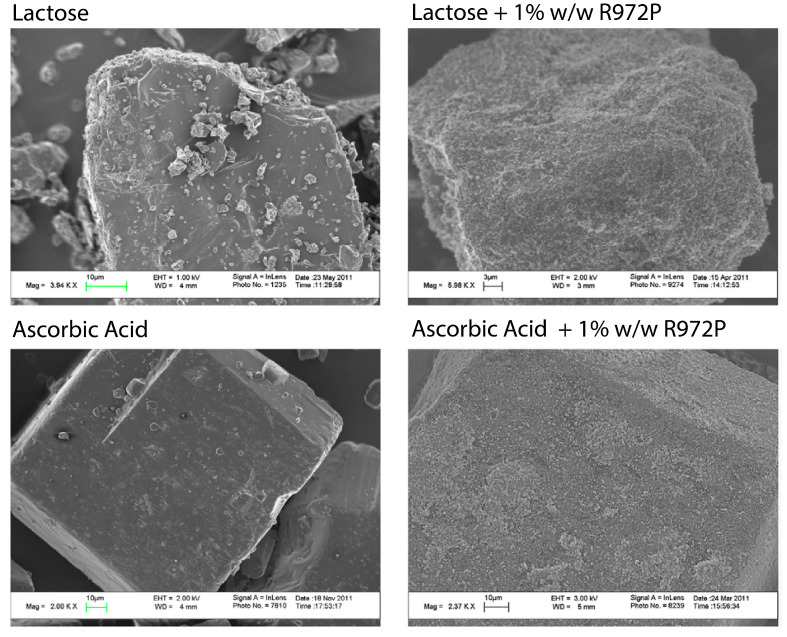
SEM images showing the surface coatings of (**top**): excipient (lactose) and (**bottom**): API (Ascorbic Acid). (**Left**): Non-coated and (**right**): 1% *w*/*w* Aerosil^®^ R972 colloidal silica coating. Adapted from Huang et al. [70].

**Figure 20 pharmaceutics-15-01587-f020:**
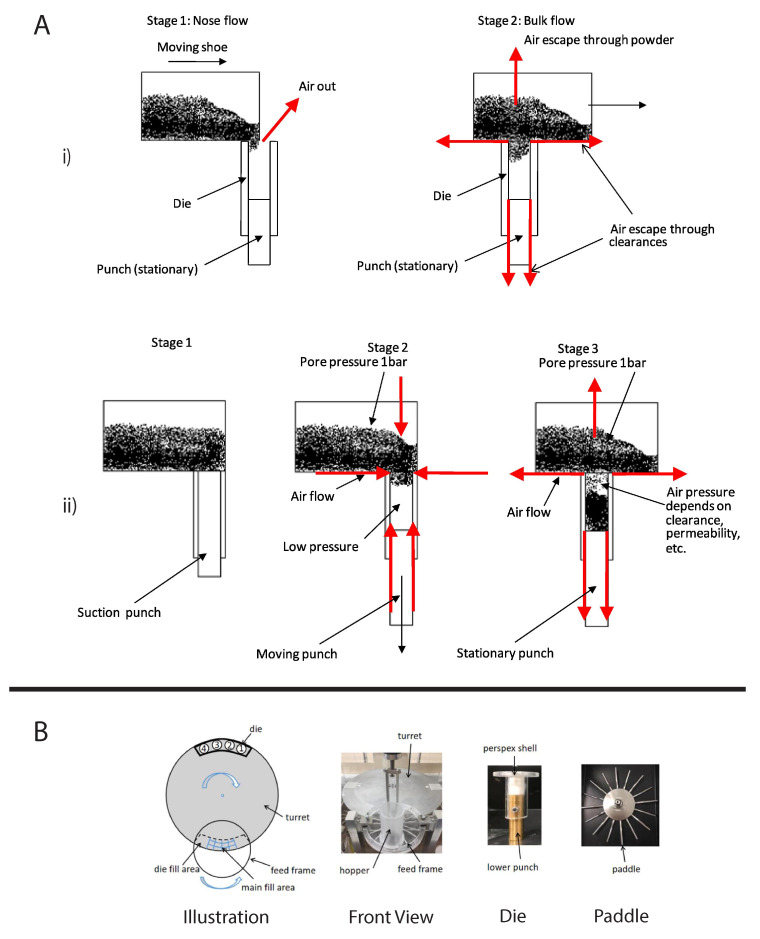
(**A**) Schematic diagram of air flow mechanism in (**i**) gravitational die-filling mechanism (**ii**) suction fill mechanism in a shoe and die system obtained from Mills and Sinka [38]. (**B**) Schematic of a rotary die-filling system, a type of force feeder, typically found in a turret tabletting machine obtained from Tang et al. [105].

**Figure 21 pharmaceutics-15-01587-f021:**
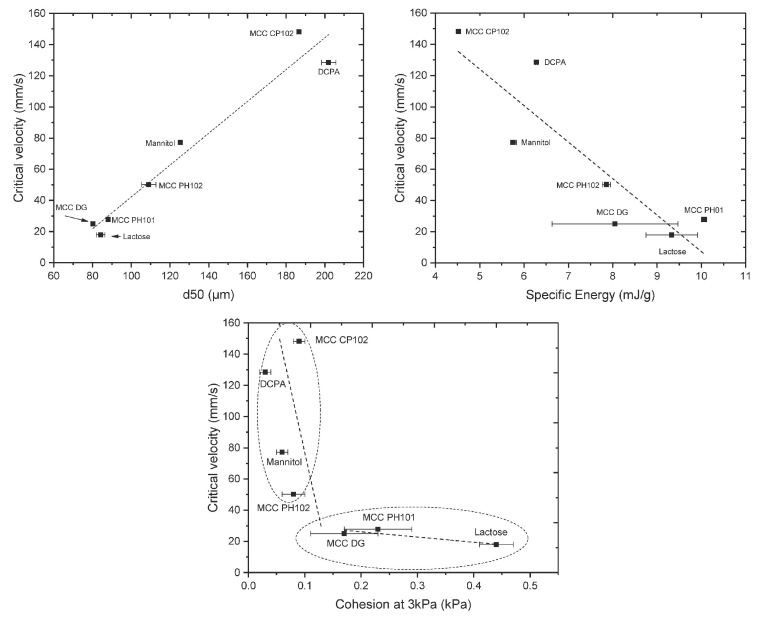
Graphs obtained from Zakhvatayeva et al. [101] showing the change of critical velocity as a function of: average particle size (**top left**), specific energy (**top right**) and cohesion at 3 kPa consolidation stress measured using the FT4 (**bottom**) of a variety of different materials.

**Figure 22 pharmaceutics-15-01587-f022:**
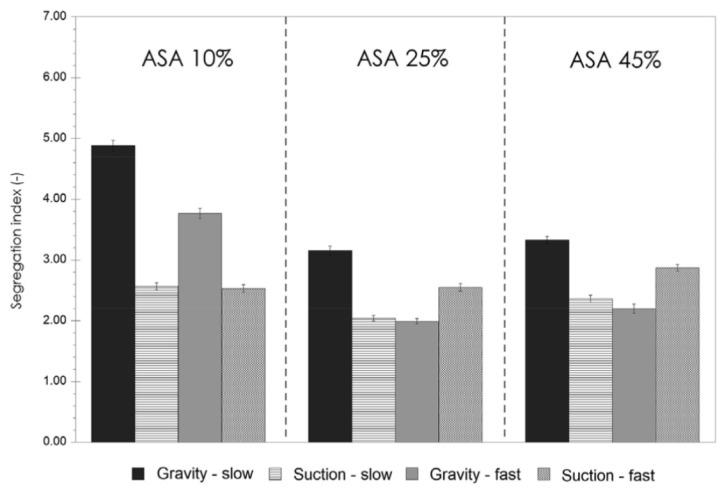
Segregation index of three different blends of Acetylsalicylic acid (ASA) with mannitol using gravity and suction filling where suction filling mostly has a lower segregation index compared to gravity from Zakhvatayeva et al. [109] the fast scenario is where the die-filling velocity is 260 mm/s and slow has a die-filling velocity of 70 mm/s.

**Figure 23 pharmaceutics-15-01587-f023:**
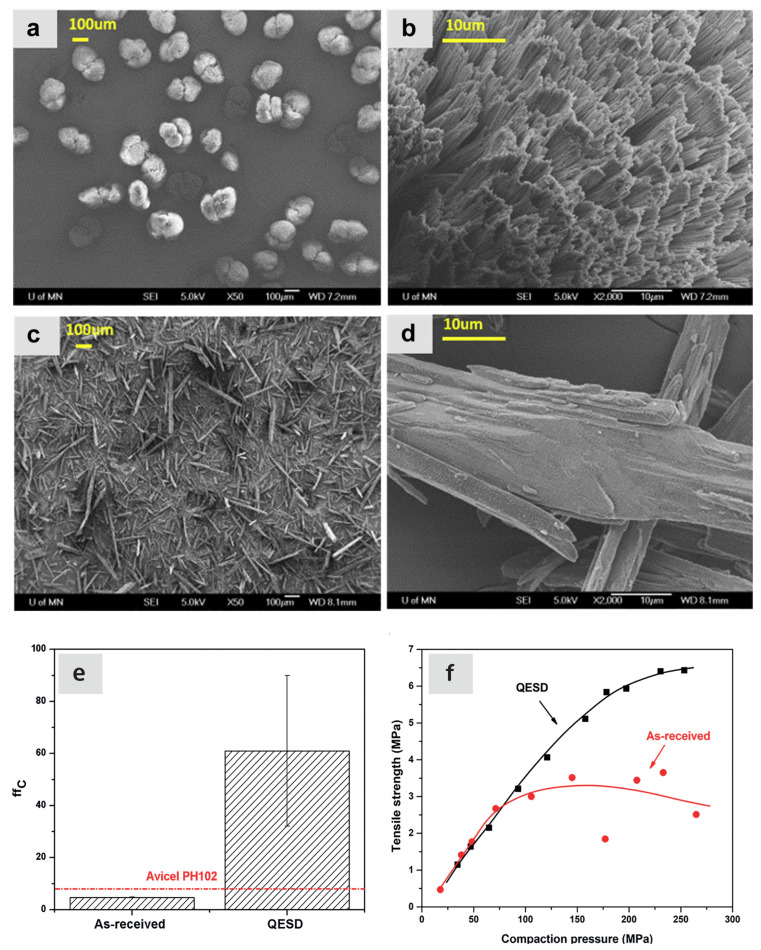
(**a**–**d**): Scanning electron microscope images of ferulic acid (FA) particles (**a**) Quasi-emulsion solvent diffusion (QESD) at 50× magnification, (**b**) QESD powder at 2000× magnification, (**c**) as-received FA powder at 50× magnification, and (**d**) as-received FA powder at 2000× magnification. (**e**,**f**): Figures to show improved tabletability and flowability with spherical agglomerated/crystallised (QESD) ferulic acid obtained from Chen et al. [18].

**Figure 24 pharmaceutics-15-01587-f024:**
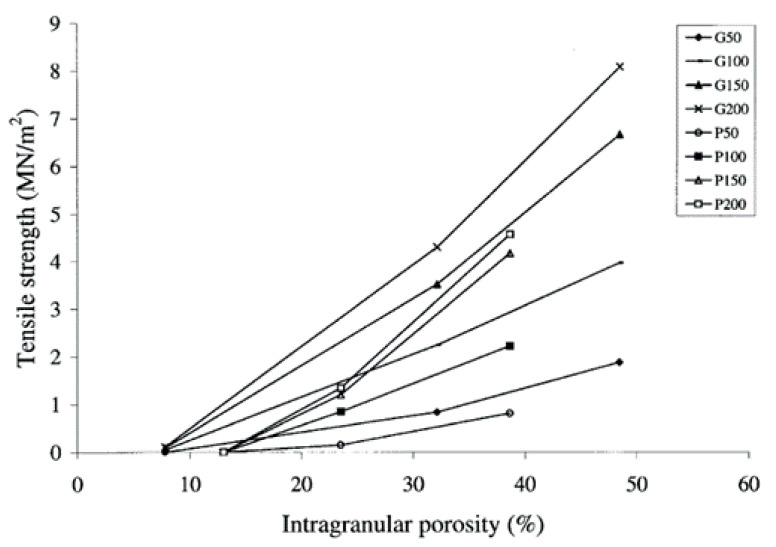
Johansson and Alderborn [30] G = granules (irregular), P = pellets (nearly spherical) the number is the compressional pressure it was compressed to e.g., G200 = granules compressed to 200 MPa.

**Figure 25 pharmaceutics-15-01587-f025:**
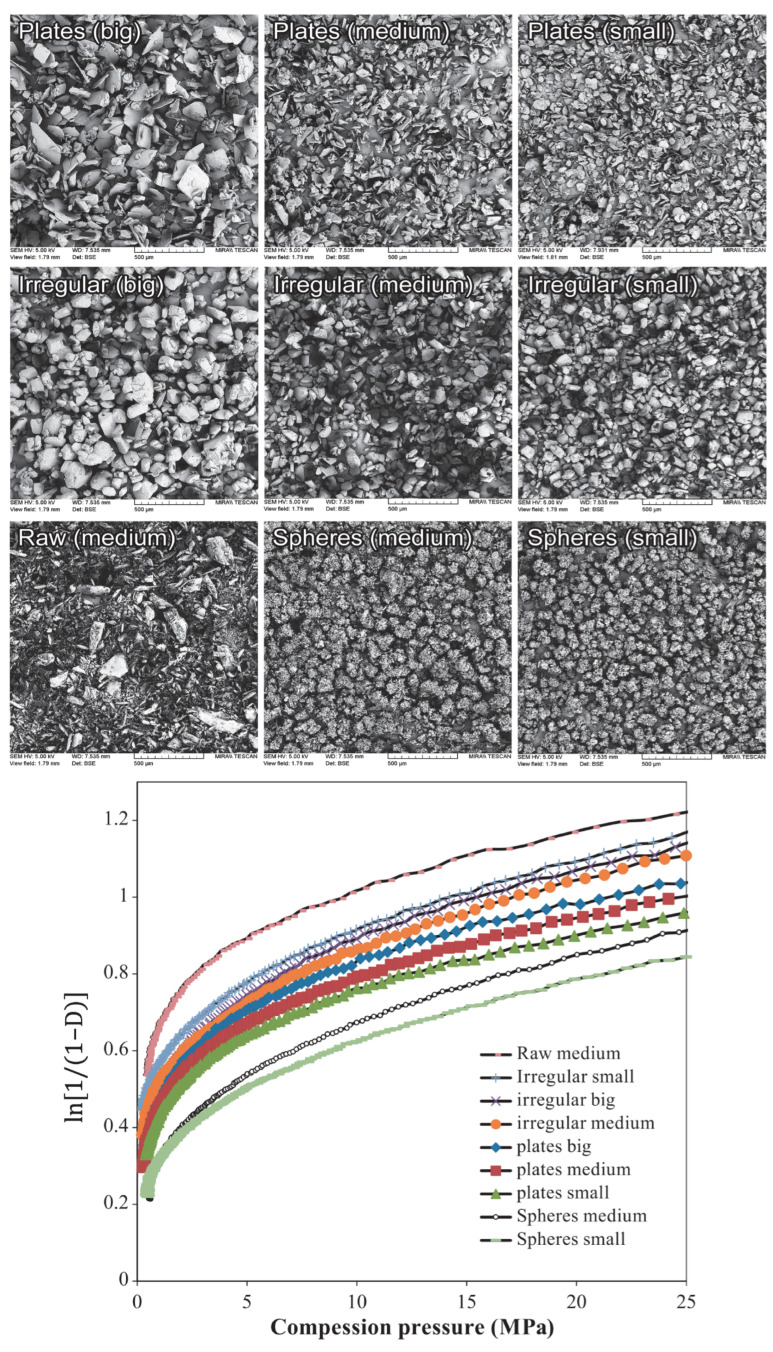
(**Top**): SEM micrographs of the paracetamol samples made (prepared and raw). The prepared samples were dissolved in ethanol and mixed at a set RPM to achieve the modified shapes. (**Bottom**): Heckel plot of different sizes and shapes of paracetamol. Both figures obtained from Šimek et al. [133].

**Figure 26 pharmaceutics-15-01587-f026:**
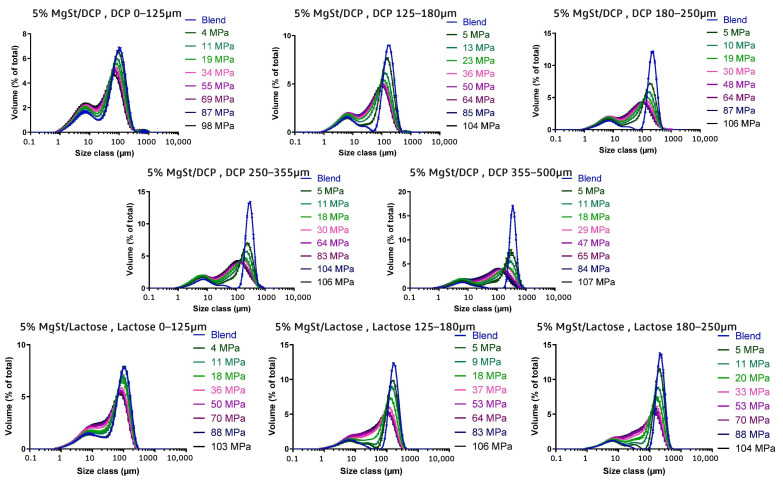
Skelbaek-Pedersen [137] PSD of calcium hydrogen phosphate dihydrate (**top**) and lactose (**bottom**) with different initial particle size distribution in the graph titles. The plots show the change in PSD with increasing levels of compressional pressure.

**Figure 27 pharmaceutics-15-01587-f027:**
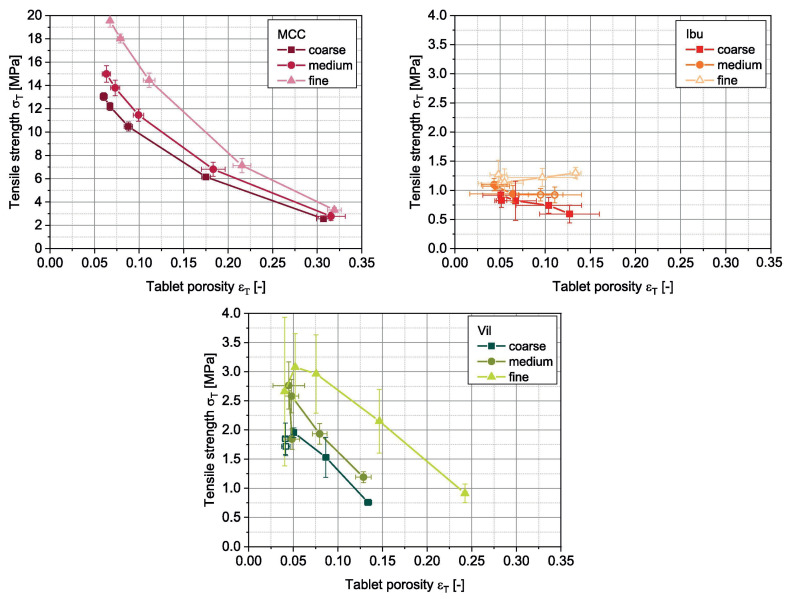
Compactability graphs of coarse, medium and fine grades of MCC, Ibuprofen and vildagliptin showing the change of compactability with particle size, obtained from Wunsch [132].

**Figure 28 pharmaceutics-15-01587-f028:**
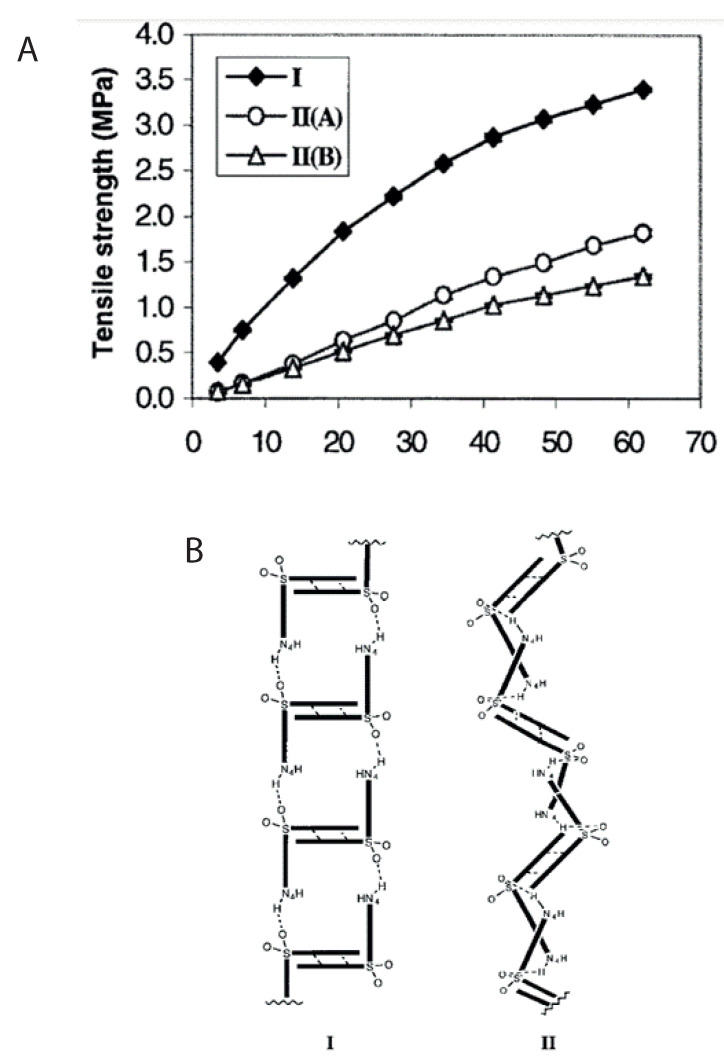
(**A**) Tabletability curves showing the difference of tabletability of three polymorphs of sulfamerazine I, II(A) and II(B). (**B**) Schematic of crystal structure of polymorphs: I and II. Hydrogen bonding are the broken lines. These crystal structure diagrams show that, even though the hydrogen bonding connectivity is the same for both polymorphs, the secondary structures are different. All figures obtained from Sun and Grant [130].

**Figure 29 pharmaceutics-15-01587-f029:**
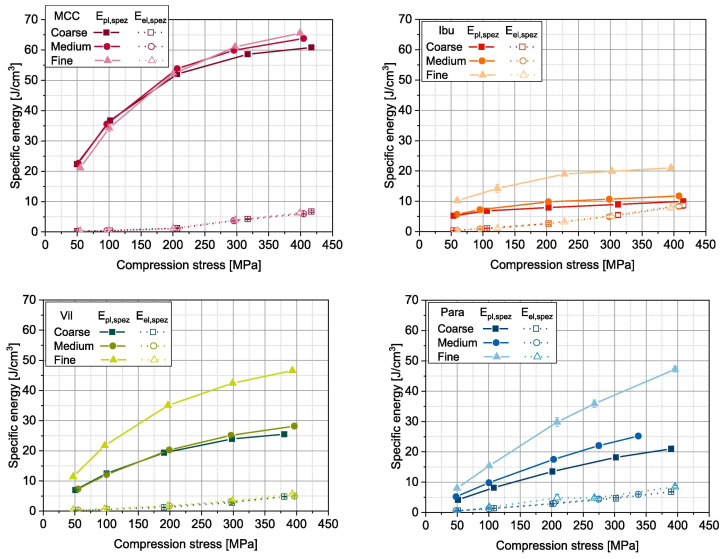
Graphs to show the specific plastic (solid lines) and elastic energies (dotted lines) of fine, medium and coarse grades of MCC, ibuprofen (Ibu) and vildagliptin (Vil) and paracetamol (para) obtained from Wunsch et al. [132] These graphs show that coarse grades have lower specific surface energy across all cases.

**Figure 30 pharmaceutics-15-01587-f030:**
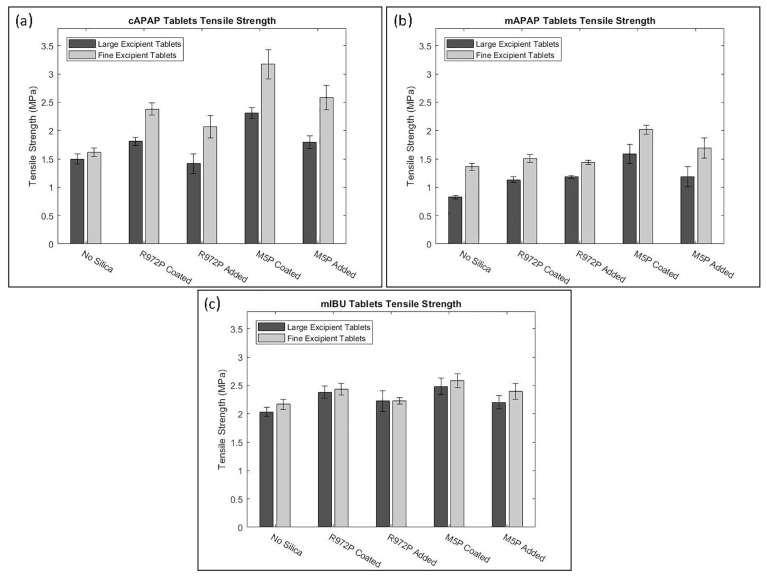
Tensile strength of the tablets made from (**a**) coarse acetaminophen (cAPAP), (**b**) micronised acetaminophen (mAPAP), and (**c**) micronised ibuprofen (mIBU) where Aerosil R972P nano-silica (hydrophobic) and CAB-O-SIL M5P nano silica (hydrophilic) obtained from Kunnath et al. [151].

**Table 1 pharmaceutics-15-01587-t001:** Material properties of the APAP grades used in Palmer et al. [75].

Material	FFC	Bulk Density (g cm${}^{-3}$)	Tapped Density (g cm${}^{-3}$)	Hausner Ratio	Carr’s Index
Micronized APAP	1.4	0.19	0.30	1.58	36.67
Powdered APAP	1.9	0.31	0.53	1.71	41.51
Special Granular APAP	19.4	0.73	0.83	1.14	12.05

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
