# Peer review of "Reviewing the Impact of Powder Cohesion on Continuous Direct Compression (CDC) Performance"

_pharmaceutics, 2023, doi:10.3390/pharmaceutics15061587_

Round 1

Reviewer 1 Report

The manuscript can be accepted in its present form. It is well written, organized, scientifically sound and informative.

I suggest the authors to give another thorough read to fix some typos that I noticed. Secondly, the punctuation is off in some places. 

Reviewer 2 Report

Although there ar no new data in this manuscript, it fulfils the need of reviewing the covered field and it was missing from the literature. I like the fact that the review is not a simple juxtaposition of the available information. It is made under a critical umbrella and the information synthesis is sound and provides new insight.

My main question to be answered before accepting the manuscript for publication is: why shedding lignt only on silication and not covering other excipients' cases? Moreover, I suggest a lengthier conclusion detailing the main synthesis of the information otop of exposing the areas of information lack and the needs for further studies.

There are some typing errors but in general the English is good. I would suggest a proofreading from a professional.

Reviewer 3 Report

The paper by Jones-Salkey et al. addresses an important topic of pharmaceutical unit operations relevant for the development of continuous manufacturing processes. The review is thorough with sufficient number of details, which are supported by 148 references, which are reasonably selected. Most references are taken from generally available resources and DOIs are provided. There are 6 references, which are hard to locate - specifically references 18, 31, 53, 90, 141, 147 are incomplete and it would be very hard to identify correct resource. It should be checked and completed with at least with publisher or URL info if possible.

In the introduction section, the cohesion principle is briefly explained and cohesion inducing factors are listed. Here Liquid bridges are mentioned as a factor that will not be discussed due to the availability of other reviews, but in my opinion there should be a differentiation between the liquid bridges, where the liquid water is present and the effects of moisture – adsorbed water, which can significantly affect cohesion due to alteration of surface energy of the particles.

Also, in the introduction and later section, the necessity of excipient selection trade-off between the requirement of low cohesivity for flow and high cohesivity for compression is mentioned (eg. Line 55). I think the existence of modern materials having combined positive effect like magnesium aluminosilicates or tricalcium phosphates could be also mentioned here.

I think that discussion of differences of colloidal silica action on different kind of excipients around Fig. 2 can benefit from the findings of surface roughness role on distributing the nanoparticles on carrier surface over time discussed in Diem Trang T., et al.: Int. J. Pharm. 556, 383 (2019).

I think the paper is ready for publication in Pharmaceutics with possible minor amendments listed above.

Reviewer 4 Report

This review provides an overview of the role of powder cohesion in the steps of continuous direct compaction, which is a hot topic as continuous manufacturing is spreading in the pharmaceutical industry. In batch technologies, the powder property of being cohesive and sticky is important, however, it becomes extremely significant in continuous operation, influencing flowability and the compatibility of the powders. The manuscript is well-written, and it highly contributes to getting a picture of the possible issues if some materials have a tendency to stick with each other or with the elements of the equipment. The processed literature is up-to-date and includes all the relevant papers. Table 2 is an excellent summary of this thorough work. Some minor issues should be clarified before acceptance for publication in this journal.

Comments:

1.       The structure of the manuscript is not the best. The splitting into sections follows the main operation steps (feeding, blending and tabletting) of CDC. The first two are fine, as feeding and blending have parts of introduction, discussion, and summary. However, in the case of tableting is not clear why the 4.3 Discussion follows the 4.2 Compression. I think Section 4.3 belongs to Compression, therefore 4.3.1 and 4.3.2 should be 4.2.1 and 4.2.2, respectively, with the following titles: e.g. Effects of Particle Size and Shape, Effects of Surface Energy

2.       What does ‘performance of tablet’ mean in Line 84?

3.       The dampening effect of RTD of the processes should be considered in this case. Not every perturbation causes problems in content uniformity. Continuous measurement with funnel plots can help to identify the more problematic disturbances in feeding. (Gyürkés et al. DOI: 10.3390/pharmaceutics12111119)

4.       In Figs 4 and 28, the use of left, middle and right is not clear as the subplots are marked with letters.

5.       Line 727 What about tablet ejection? Should there not be a separate section for this? For example, before Section 4.4. Or the cohesion does not affect ejection force. Or can lubricants handle it?

6.       In abbreviations, the ‘rpm’ is known as Revolutions Per Minute.

7.       Please use abbreviated journal names in reference list.

8.       Some references are not complete (e.g. Refs. 53, 125 and 147)

English language is fine. The

Round 2

Reviewer 2 Report

All requested correction and explanation have been addressed satisfactorily by the authors.